



# SEA-Rice-Ci10: High-resolution Mapping of Rice Cropping Intensity and Harvested Area Across Southeast Asia using the Integration of Sentinel-1 and Sentinel-2 Data

Frisa Irawan Ginting[1], Rudiyanto[1,8]*, Fatchurrachman[1], Ramisah Mohd Shah[1], Norhidayah Che Soh[1],
Sunny Goh Eng Giap[2], Dian Fiantis[3], Budi Indra Setiawan[4], Sam Schiller[5,8], Aaron Davitt[6,8], Budiman
Minasny[7]*

[1]Program of Crop Science, Faculty of Fisheries and Food Science, Universiti Malaysia Terengganu, Kuala Nerus 21030,
Terengganu, Malaysia
[2]Program of Environmental Technology, Faculty of Ocean Engineering Technology and Informatics, Universiti Malaysia
Terengganu, 21030, Kuala Nerus, Terengganu, Malaysia
[3]Department of Soil Science, Faculty of Agriculture, Andalas University, Kampus Limau Manis, Padang 25163, Indonesia
[4]Department of Civil and Environmental Engineering, IPB University, Bogor, 16680, Indonesia
[5]Carbon Yield, Chicago IL, 60660 USA
[6]WattTime, Oakland CA, 94609 USA
[7]School of Life and Environmental Sciences, The University of Sydney, 1 Central Avenue, The Australian Technology Park,
Eveleigh, NSW 2015, Australia
[8]Climate TRACE, Global coalition – USA and Malaysia

*Correspondence to*: Rudiyanto (rudiyanto@umt.edu.my) and Budiman Minasny (budiman.minasny@sydney.edu.au)

**Abstract.** The Southeast Asia region has a vast expanse with diverse tropical climates, making it a prominent centre of rice cultivation, contributing to about 20% of the world's rice production and contributes 29% of global rice methane emissions. As a staple food for many countries, accurate and up-to-date information on the rice harvested area is crucial for addressing food security issues, predicting rice yield and methane emissions, and formulating effective government policies. This paper presents the first detailed study of rice cropping intensity and harvested areas throughout Southeast Asia. Current remote sensing products have not been able to produce up-to-date cropping intensity information due to the variability of local cultivation practices. To address this problem, we integrated Sentinel-1A and Sentinel-2A/B time series data from 2020 to 2021 and developed a local unsupervised classification with phenological labelling (LUCK-PALM) method. We implemented the system on the Google Earth Engine (GEE) cloud-based platform to produce geospatial products of rice cropping intensity and harvested area at a spatial resolution of 10 m called SEA-Rice-Ci10s. The results show that Southeast Asia's total rice growing area in 2020-2021 was 28.5 Mha, with 51% single cropping, 47% double cropping, and 2% triple cropping. These were equivalent to 42.9 Mha of annual harvested area, consisting of Thailand (11.2 Mha), Indonesia (8.4 Mha), Myanmar (8.4 Mha), Vietnam (6.3 Mha), Cambodia (3.9 Mha), the Philippines (3.3 Mha), Laos (0.8 Mha), Malaysia (< 0.5 Mha), and Timor-Leste (0.01 Mha). We compared our rice maps to agricultural statistics data at the district and province levels and existing rice maps for some Southeast Asian countries. The results demonstrate that our map agreed well with countries' statistics with a linear coefficient of determination ($R^2$) from 0.85 to 0.97. Compared to existing products, our map can resolve small paddy





fields of about 0.2 ha in the hilly areas. This information will be useful in addressing food security challenges and improving estimates of methane emissions from rice cultivation. The 10 m paddy rice cropping intensity map for Southeast Asia, SEA-Rice-Ci10, is available on the GEE App (https://rudiyanto.users.earthengine.app/view/seariceci2021), the Climate TRACE platform (https://climatetrace.org/) and the Zenodo repository (https://doi.org/10.5281/zenodo.10707621) (Frisa Irawan et al., 2024).

**1 Introduction**

Accurate and up-to-date information on the harvested area of rice is crucial for addressing food security issues, identifying and predicting rice yield, managing water resources, calculating methane emissions ($CH_4$) released into the atmosphere, and formulating effective government policies (IPCC, 2019; Chandra Paul et al., 2020; Karthikeyan et al., 2020). However, the data on harvested rice area, typically obtained from agricultural statistics, can be notably outdated, often reflecting figures from several years prior. Additionally, this information is usually aggregated at broader levels: either at the national level for The Food and Agriculture Organization Corporate Statistical Database (FAOSTAT) or at administrative levels one (province or state) and two (district) for national agricultural statistics. While informative, neither of these sources provide the detailed spatial information required to tackle the issues mentioned effectively. As an alternative, over the past decade, remote sensing technologies have extensively advanced in mapping the extent of rice fields. These technologies focus primarily on mapping rice extent or growing areas (i.e., areas cultivated with paddy at any point during a given year) (Xiao et al., 2006; Chen et al., 2020; Sun et al., 2023). However, they fall short of accurately mapping the total rice harvested area, which includes multiple seasons in a year.

In the Southeast Asia region, temperature and water availability do not limit the rice growing period, and thus, rice can be grown throughout the year with one, two, or even three harvests (Rudiyanto et al., 2019; Fatchurrachman et al., 2022). Summing individual rice fields in Southeast Asia does not equate to the total harvested area, unlike in temperate regions such as China, India, the USA, Italy, and Japan where rice paddies typically produce one harvest a season. This presents a challenge for current remote sensing methods as the rice cropping calendar varies throughout Southeast Asia.

Mapping rice areas nationally and globally usually employs remote sensing data from medium to high-spatial resolution optical and radar sensors. It includes optical (e.g., MODIS, Landsat, Sentinel-2, and SPOT) and Synthetic Aperture Radar (SAR) (e.g., RADARSAT, ALOS PALSAR, and Sentinel-1) to identify and map rice areas extent (Clauss et al., 2018; Bazzi et al., 2019; Rudiyanto et al., 2019). Sentinel-1 SAR is particularly effective in tropical regions due to its cloud penetrating capabilities and ability to detect rice growth due to SAR's unique backscatter response caused by initial rice stem growth in a flooded field (Nelson et al., 2014; Singha et al., 2019; Davitt et al., 2020). However, the data contains speckle noise, which can reduce the accuracy. Conversely, the optical Sentinel-2 sensor excels in identifying unique spectral characteristics of a crop at different growth stages (Zhao et al., 2021), however, its effectiveness is limited by cloud cover. The



integration of Sentinel-1 and 2 data has proven to be beneficial in improving rice mapping accuracy, offering high spatial and temporal resolution (10 m over 5-12 days) (Ramadhani et al., 2020; Han et al., 2021; Fatchurrachman et al., 2022). These sensors are crucial for more frequent mapping of small and hilly rice fields in Southeast Asia, which help identify the varying rice cropping calendars in Southeast Asia (Ramadhani et al., 2020; Han et al., 2021; Fatchurrachman et al., 2022).

Methods for mapping rice fields using remote sensing include machine learning classifiers (e.g., random forest and support vector machines) (Onojeghuo et al., 2018; Lakshmanaprabu et al., 2019; Ramadhani et al., 2020; Zhang et al., 2020b; Thorp and Drajat, 2021), expert phenology-based classifiers (He et al., 2018; Ding et al., 2020; dela Torre et al., 2021; Fatchurrachman et al., 2022; Huang et al., 2023). Supervised machine learning classification methods require extensive training data, creating challenges in collecting representative samples, particularly throughout Southeast Asia. Conversely, expert phenology-based classifiers are straightforward, albeit requiring in-depth expertise for labelling time series of vegetation indices for rice growth stages (Rudiyanto et al., 2019; Han et al., 2021; Fatchurrachman et al., 2022).

Studies on mapping rice cropping areas have employed various resolutions, ranging from 250 m to 500 m globally, and finer resolutions at 10 m in regional studies (Laborte et al., 2017; Rudiyanto et al., 2019; Mishra et al., 2021; Fatchurrachman et al., 2022; Zhao et al., 2023). Despite these advancements, the variable local cultivation practices in Southeast Asia present a significant challenge. For the Southeast Asia region, annual paddy rice planting area and cropping intensity maps are only available at a coarser resolution (500 m) for Asia, such as Han et al. 2022. At finer resolution (10-20 m), only maps of rice-growing areas are available, e.g., Han et al. 2021. Supervised classification methods do not have sufficient training data to cover the variability of rice practices across the region. Similarly, phenological methods are challenged by vegetation indices and other indicators that vary throughout the regions. Thus, mapping rice cropping intensity and harvested areas across Southeast Asia at a fine spatial resolution remains an unmet need.

This study aims to generate a geospatial product of rice cropping intensity at a 10 m spatial resolution and harvested area by integrating Sentinel-1 and Sentinel-2 time series data from 2020 to 2021 across Southeast Asia countries. To address the challenges of varying local practices, we introduce the local unsupervised classification with phenological labelling (LUCK-PALM) approach, integrating unsupervised k-means clustering and expert phenological labelling within local areas. The algorithm was implemented in Google Earth Engine (GEE) platform. This study is the first to report on a high-spatial resolution map (10 m) of rice cropping intensity and harvested area across Southeast Asia for 2020 to 2021. The outcome of this product is anticipated to be instrumental in addressing food security challenges and addressing environmental concerns, particularly in estimating methane gas emissions associated with rice cultivation.

## 2 Materials and methods

### 2.1 Study area

Southeast Asia comprises 11 countries located between latitude −10°–30°N and longitude 90°–141°E with a total area of 450 Mha (Fig. 1a). The study area focused on nine countries in Southeast Asia, comprised of Indonesia, Timor-Leste, Malaysia,



Thailand, Cambodia, Vietnam, Laos, the Philippines, and Myanmar (Fig. 1). Brunei and Singapore are not included because
of their relatively small rice growing area.

Southeast Asia countries produce approximately 20% of the world's rice production (Food and Agriculture
Organization, 2024). It is a tropical climate region with abundant rainfall (Singh and Qin, 2020), with an average annual rainfall
of 2,241 mm, and an average range of 1,622 to 2,875 mm (Precipitation-Country rankings, 2024). It has an annual average
temperature of 25 ºC, varying between 20 to 33 ºC (Raghavan et al., 2018). The climate according to the Koppen-Geiger
climate classification is tropical rainforest (Af) for Indonesia, Malaysia and south and east regions of the Philippines; tropical
monsoon (Am) for north and west of the Philippines, northern central and south regions of Vietnam; and tropical savannah
(Aw) for Thailand, Myanmar, Cambodia, Laos, the southern central region of Vietnam and Timor-Leste (Carver et al., 2002).
Rainfed lowland rice and irrigated lowland rice are the two main types of agriculture in Southeast Asia (Silva et al., 2022).
Two rice harvests per year are common, and in some areas with adequate irrigation, it is possible to harvest three times a year
(Fig. 1).

(a)

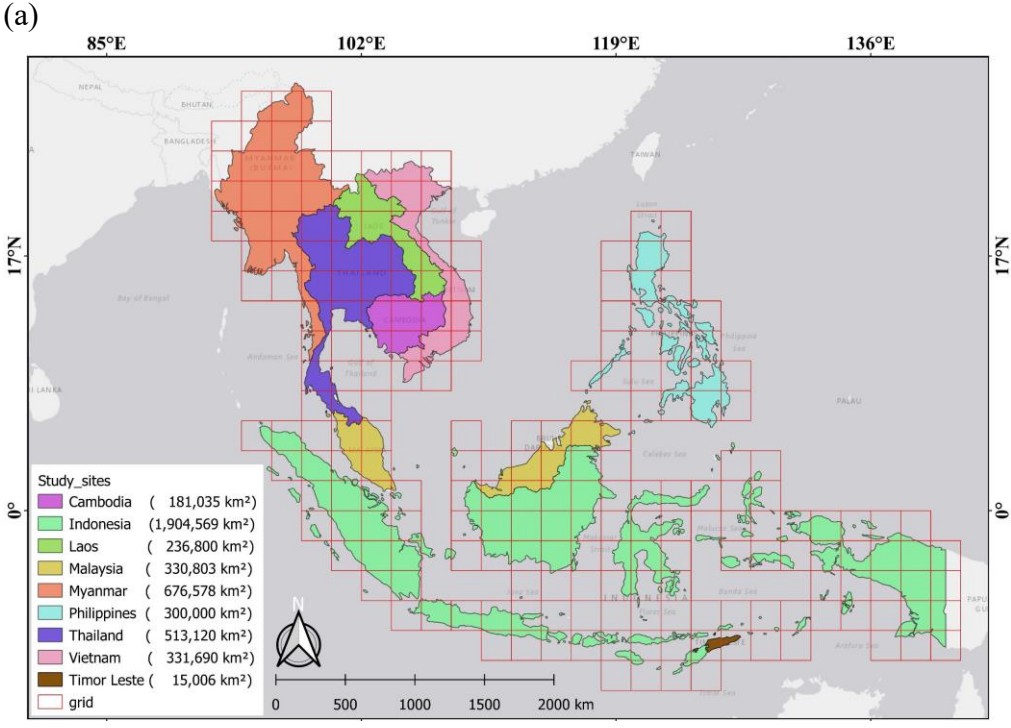



(b)

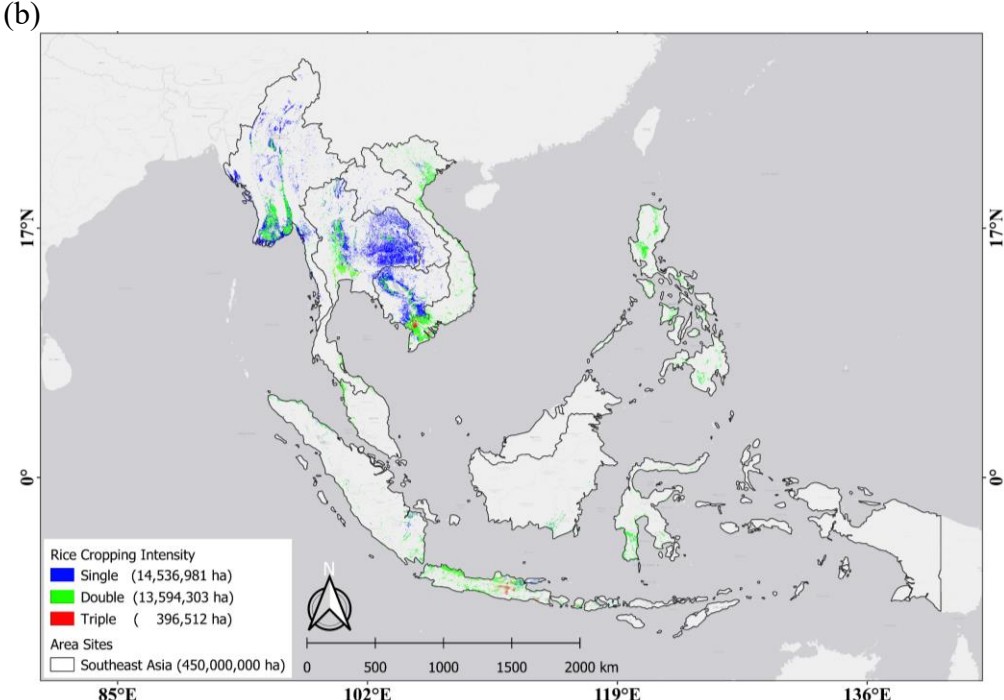

**Figure 1: (a) The study area location comprised 9 countries in Southeast Asia. Included in this figure are the grids (red squares) utilised in the local adaptive modelling and the total country area in km². (b) Distribution and cropping intensity of paddy rice for the period of 2020-2021 based on this study. Colored areas are the following: blue = single rice planting, green = double rice planting, and red = triple rice planting.**

## 2.2 Datasets

### 2.2.1 Sentinel-1 and Sentinel-2 time-series data

Sentinel-1 and Sentinel-2 time series data were obtained from January 2020 to December 2021 using the Google Earth Engine (GEE) data catalogue (https://developers.google.com/earth-engine/datasets/). A total of 31,711 scenes from Sentinel-1 and 208,197 scenes from Sentinel-2A and B were retrieved from the GEE platform. The characteristic parameters of the Sentinel-1 and Sentinel-2 data used in this study are shown in Table 1.

Sentinel-1 ground-range-detected (S1_GRD) images with instrument mode Interferometric Wide (IW) and Vertical-transmit/Horizontal-receive (VH) band that are available on the GEE were utilised as the primary data source for our paddy field mapping method because the C-band SAR can obtain data through cloud cover (e.g., Davitt et al., 2020). This study only used VH-polarized backscatter data because VV backscatter is less responsive to rice fields (Rudiyanto et al., 2019; Fatchurrachman et al., 2022; Xu et al., 2023). GEE pre-processed the data with the Sentinel-1 Toolbox to remove thermal noise, radiometric calibration, orbital correction, and terrain correction (https://developers.google.com/earth-engine/guides/sentinel1).





Sentinel-2 (referred to Sentinel 2A/B) Level-2A (Surface Reflectance, SR) images have many artefacts and overcorrected and therefore were substituted with Sentinel-2 Level-1C (Top-of-Atmosphere Reflectance, TOA) (Brinkhoff et al., 2022;
Fatchurchman et al., 2022). More information on the Sentinel-2 MSI product can be found in the User Guides on the ESA website (https://sentinel.esa.int/web/sentinel/user-guides/sentinel-2-msi, accessed on 7 September 2023). This study used four bands of Sentinel-2: band 4 (Red) and band 8 (Near-infrared or NIR) which were used to calculate Normalised Difference Vegetation Index (NDVI); and band 3 (Green) and band 11 (Shortwave infrared 1 or SWIR), which were used to generate Modified Normalized Difference Water Index (MNDWI). The application of Sentinel-1 and -2A/B measurements is discussed
in section 2.3 Methodology.

**Table 1.** Specifics of the Sentinel-1 and Sentinel-2 data obtained from GEE

| Sentinel-1 (12 Days Revisit Time) | | | | Sentinel-2 (5 Days Revisit Time) | | | |
|---|---|---|---|---|---|---|---|
| Band | Resolution (m) | Sensor | Polarization | Band | Wavelength (nm) | Resolution (m) | Sensor |
| VH | 10 | C-band SAR | VH (IW Mode) | B3 - Green | 560 | 10 | |
| | | | | B4 - Red | 665 | 10 | |
| | | | | B8 - Near-infrared | 842 | 10 | MSI |
| | | | | B11 - Shortwave infrared 1 | 1610 | 20 | |

### 2.2.2 Agricultural statistics

Our remote sensing-derived rice intensity and mapping data were compared to the agricultural statistics yearbooks of each nation were collected to compile annual census data of rice growing areas (i.e., parcel areas) and rice harvested areas at various administrative levels in these countries (Table 2). The administrative levels include national and subnational levels (state, province, or regions, uniformly represented by province in this study). The unit of area in the statistical yearbook was converted to ha.

### 2.2.3 Existing Paddy Rice Maps

Two rice field maps with the same area coverage and similar spatial resolution were compared with our results (Table 2). The first map was produced by Han et al. (2021) using a combination of MODIS and Sentinel-1 images collected from 2017 to 2019, downloaded from https://doi.org/10.5281/zenodo.5645344. This map, referred to as the "NESEA-Rice10", covers monsoon Asia, including Southeast Asia and Northeast Asia, with a spatial resolution of 10 meters. The second map was
created using Sentinel-1 data collected in 2019 by Sun et al. (2023) for mainland Southeast Asia, and the data was downloaded from https://doi.org/10.5281/zenodo.7315076 and is referred as "20mRice-MSEAsia". Both maps present rice-growing areas rather than rice cropping intensity or harvest areas.





**Table 2.** Specifics of the data used in this study. The year column represents year of data acquisition or collection and the resolution or administrator level represents the spatial scale where Level 0 = nation or country level of administration, Level 1 = province or state level of administration and Level 2 = regency or district level of administration

| Data type | Data product or country name | Year | Resolution or Administrator Level | Data specification | Data access | Last access |
|---|---|---|---|---|---|---|
| Satellites Imagery | Sentinel-1A | 2020-2021 | 10 m | SAR GRD: C-band Synthetic Aperture Radar Ground Range Detected Level-1C, Top-of-Atmosphere Reflectance | https://developers.google.com/earth-engine/datasets/catalog/COPERNICUS_S1_GRD | 2023-09-07 |
| | Sentinel-2A and B | 2020-2021 | 10 m | | https://developers.google.com/earth-engine/datasets/catalog/COPERNICUS_S2 | 2023-09-07 |
| Crop land | ESA WorldCover 10m v100 | 2020 | 10 m | v100 | https://developers.google.com/earth-engine/datasets/catalog/ESA_WorldCover_v100 | 2023-09-07 |
| Statistics | Indonesia | 2021-2022 | Level 2 | Growing area | http://scs1.litbang.pertanian.go.id/ | 2023-06-09 |
| | Philippines | 2021-2022 | Level 2 | Growing area | https://prism.philrice.gov.ph/dataproducts/ | 2023-06-09 |
| | Malaysia | 2022 | Level 1 | Growing and harvested area | http://www.doa.gov.my/index.php/pages/view/622?mid=239 | 2023-06-09 |
| | Vietnam | 2022 | Level 1 | Harvested area | https://www.gso.gov.vn/en/data-and-statistics/2023/06/statistical-yearbook-of-2022/ | 2023-06-09 |
| | Thailand | 2021 | Level 1 | Harvested area | http://statbbi.nso.go.th/staticreport/page/sector/en/11.aspx | 2023-06-09 |
| | Cambodia | 2019 | Level 1 | Harvested area | http://nis.gov.kh/index.php/km/12-publications/15-agriculture-census-in-cambodia-2013-final-result | 2023-06-09 |
| | Laos | 2021 | Level 1 | Harvested area | https://laosis.lsb.gov.la/tblInfo/TblInfoList.do | 2023-06-09 |
| | Myanmar | 2020 | Level 1 | Harvested area | https://www.csostat.gov.mm/Content/PublicationAndRelease/2021/9.htm | 2023-06-09 |
| | Timor-Leste | 2019 | Level 0 | Growing area | http://timor-leste.gov.tl/?p=26100&lang=en | 2024-01-24 |



| Data type | Data product or country name | Year | Resolution or Administrator Level | Data specification | Data access | Last access |
|---|---|---|---|---|---|---|
| Existing rice maps | Southeast Asia countries | 2021 | Level 0 | Harvested area | https://www.fao.org/faostat/en/#data/GCE | 2024-01-28 |
| | NESEA-Rice10 | 2019 | 10 m | Growing area | https://doi.org/10.5194/essd-13-5969-2021 | 2023-06-09 |
| | MSEAsia20 | 2019 | 20 m | Growing area | https://doi.org/10.5194/essd-15-1501-2023 | 2023-06-09 |

## 2.3 Methodology

We introduce the local unsupervised classification with phenological labelling (LUCK-PALM) method to produce a map of rice growing intensity (SEA-Rice-Ci10). The flowchart is illustrated in Fig. 2, and each step is explained in the sections below:

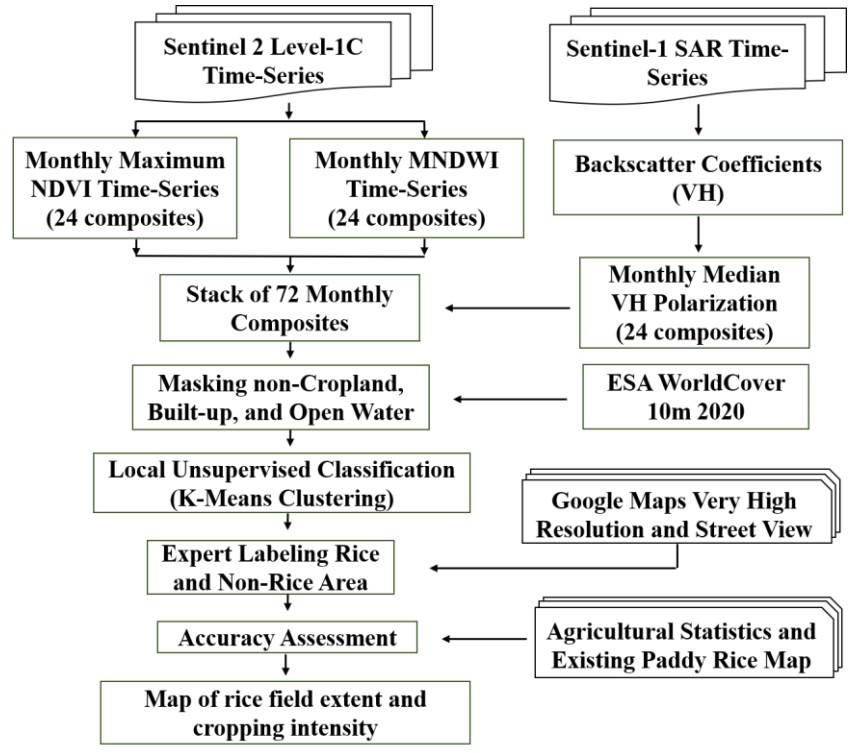


**Figure 2: Workflow of LUCK-PALM for mapping rice extent and cropping intensity using combined Sentinel-1 and Sentinel-2 time-series data.**



### 2.3.1 Monthly composite Sentinel-1 and Sentinel-2 data

LUCK-PALM uses monthly composite imagery time series data as the input dataset. The aggregated monthly data is necessary
because the two satellites, Sentinel-1 and Sentinel-2, acquired images at different times, and it also helps in reducing speckle
noise and lack of observations due to clouds. We calculated the monthly composite of Sentinel-1 VH backscatter coefficients
to obtain representative time series data of rice growth dynamics. The Median Value Composite (MedVCs) was calculated to
obtain the monthly composite of VH data because it can reduce the speckle noise, which is a random granular disturbance
present in SAR images, resulting in abrupt spikes or drops in pixel-based time series in the VH backscatter, especially in areas
with overlapping data scenes (Rudiyanto et al., 2019; Fatchurrachman et al., 2022). Numerous studies have reported that the
monthly median composite is also very effective in detecting the presence of water during transplanting, a method practically
used in rice cultivation in Southeast Asian countries, as well as in rice phenology identification (Rudiyanto et al., 2019;
Fatchurrachman et al., 2022). A total of 24 monthly median composite imagery data (for 2020-2021) were obtained.

To capture rice phenology, NDVI from Sentinel-2 was used in identifying rice growth stages and differentiating rice
plants from other land covers, while MNDWI from Sentinel-2 identified flooded areas during rice transplanting (Sakamoto et
al., 2018; de Lima et al., 2021). MNDWI was used to improve the detection of flooded areas while reducing the prominence
of built-up features, which often exhibit similar reflectance characteristics to water or wet surfaces (Mansaray et al., 2019).
NDVI was computed based on TOA reflectance band 4 (red) and band 8 (NIR) and MNDWI from the TOA reflectance band
3 (green) and band 11 (SWIR) of Sentinel-2 as follows:

$$NDVI = \frac{\text{NIR (band 8)} - Red\ (band\ 4)}{\text{NIR (band 8)} + Red\ (band\ 4)} \tag{1}$$

$$MNDWI = \frac{\text{Green (band 3)} - SWIR\ (band\ 11)}{\text{Green (band 3)} + SWIR\ (band\ 11)} \tag{2}$$

Monthly composites for NDVI and MNDWI time series data were generated using the monthly maximum value
composites (MaxVCs). MaxVCs effectively reduce image exposure, cloud cover, and variations in atmospheric constituents,
particularly deviations in water vapour and aerosol concentrations (Holben, 1986). Several studies have effectively used
MaxVCs with MODIS, Landsat, Sentinel and SPOT data (Gumma, 2011; de Bie et al., 2012; Nguyen et al., 2012; Coulter et
al., 2016; Fatchurrachman et al., 2022). In total, 24 monthly composites for NDVI and 24 monthly composites for MNDWI
were generated from this Sentinel-2 data processing for the years 2021 to 2022.

### 2.3.2 Masking non-cropland areas

To filter out non-cropland areas in Sentinel-1 and -2 composite imagery, the European Space Agency (ESA) WorldCover 10
m 2020 product was employed. This product provides a global land cover map for 2020 at a 10 m resolution, derived from
Sentinel-1 and Sentinel-2 data (Zanaga et al., 2021). The WorldCover 2020 v100 dataset, accessible on Google Earth Engine
(GEE) with Asset ID ESA/WorldCover/v100, comprises 11 land cover categories, with an overall accuracy of 80.7% in Asia.





Land categories 10, 20, 30, 50, 60, 70, 80, 90, 95, and 100 were utilized to filter out non-croplands in each Sentinel-1 and -2 composite imagery. Waterbody, tree, and build-up layers from the WorldCover dataset were employed to mask non-cropland areas in Southeast Asia. This masking operation not only reduced computational memory and processing time but also minimized the false classification rate of rice fields. Consequently, only the Sentinel-1 and -2 composite imagery corresponding to crop areas were classified into rice and non-rice classes, enabling the assessment of rice cropping intensity.

### 2.3.3 Grid based clustering

Classifying large-scale or global land cover could be achieved through two modelling options, i.e., global classification (Teluguntla et al., 2018; Brown et al., 2022) and local classification models (Singha et al., 2016; Zhang et al., 2020c; Phalke et al., 2020). Global or whole-area classification modelling involves using all training samples to train a single classifier, which was applicable for land-cover mapping in the whole area (e.g., Buchhorn et al. (2020)). This contrasts with local classification modelling, which divides the whole area into regions, trains local classifiers with local samples, and creates a global or whole-area land-cover map by spatial mosaicking all local classification results. For instance, Zhang et al. (2021) split the world into grids of 5º x 5º and then developed corresponding local models for each grid to produce the regional land-cover results. Some advantages of local adaptive classification modelling are that the local adaptive models achieved higher accuracy performance than the single global model (Zhang and Roy, 2017), facilitated the regional adjustment of classification parameters, incorporated regional characteristics and enhanced the sensitivity of the training samples (Radoux et al., 2014). Additionally, it tackles the issues of computational capacity and memory limitations, which are common challenges in global models.

Here, we devised a sampling method with a 2° by 2° grid, segmenting Southeast Asia into 215 grids. This approach was designed to optimise computational efficiency, accuracy, and the quantity of training samples (Fig. 1a). This method also accommodates the diverse local rice cultivation calendars and their varying satellite indices response.

LUCK-PALM uses the unsupervised K-Means classification method to group stacked imagery of 72 monthly composite indices (monthly layers from January 2020 to December 2021) of VH backscatter (24 bands) from Sentinel-1 data, NDVI (24 bands), and MNDWI (24 bands) from Sentinel-2 data into rice and non-rice groups. LUCK-PALM employed K-Means classification due to its relative simplicity in implementation, capability to handle large datasets, assurance of convergence, ease of adaptation to new examples, and effective generalisation to clusters of various shapes and sizes (Philip Wong et al., 2023). This classification was implemented in the GEE platform using the "ee.Clusterer.wekaKMeans" function. In this implementation, 2,000 random points were sampled at each grid and used as training data in the K-Means model with 25 to 30 clusters output. The chosen number of clusters is assumed to represent the spectral data variations adequately. The algorithm automatically normalised numerical input attributes and used the Euclidean distance to measure distances between clusters, to minimise within-cluster variation (distances) (Rudiyanto et al., 2019; Sitokonstantinou et al., 2021).



### 2.3.4 Extracting representative spectra profiles

To identify rice or non-rice groups, it is necessary to obtain the representative spectra of VH backscatter, NDVI, and MNDWI
for each cluster generated by the K-means. To achieve this, we extracted the respective spectra from 2,000 random samples
from the defined area. Subsequently, we applied the spatial median for each cluster to obtain the representative spectra profiles
(Rudiyanto et al., 2019; Fatchurrachman et al., 2022). The representative spectra profiles of VH backscatter, NDVI, and
MNDWI for clusters of rice field, water body, tree, and built-up are illustrated in Fig. 3.

### 2.3.5 Expert labelling and identifying paddy rice fields

Rice fields have a unique time series profile in which the VH backscatter and NDVI values fluctuate seasonally and differ
from other land uses (e.g., water, trees, other crops) as shown in the representative cluster spectra profiles in Fig. 3. Spectral
time series of non-cropping areas were relatively constant. The NDVI and VH backscatter coefficients of paddy rice change
as it grows and matures. During rice transplanting or flooding phase, NDVI and VH backscatter coefficients have the lowest
value while MNDWI reaches a maximum peak. The NDVI and VH backscatter coefficients rise after transplanting as the
paddy rice grows and develops a peak at the heading stage (Davitt et al. 2020; Zhang et al., 2020b; Huang et al., 2023). After
the rice harvest, the NDVI and VH backscatter coefficients decrease (Ramadhani et al., 2020; Fatchurrachman et al., 2022).
These three spectral characteristics were used in LUCK-PALM to distinguish rice fields from other types of land cover
(Rudiyanto et al., 2019; Zhang et al., 2020; Fatchurrachman et al., 2022). Representative spectra profiles for each cluster
derived from k-means clustering were labelled based on their phenology.




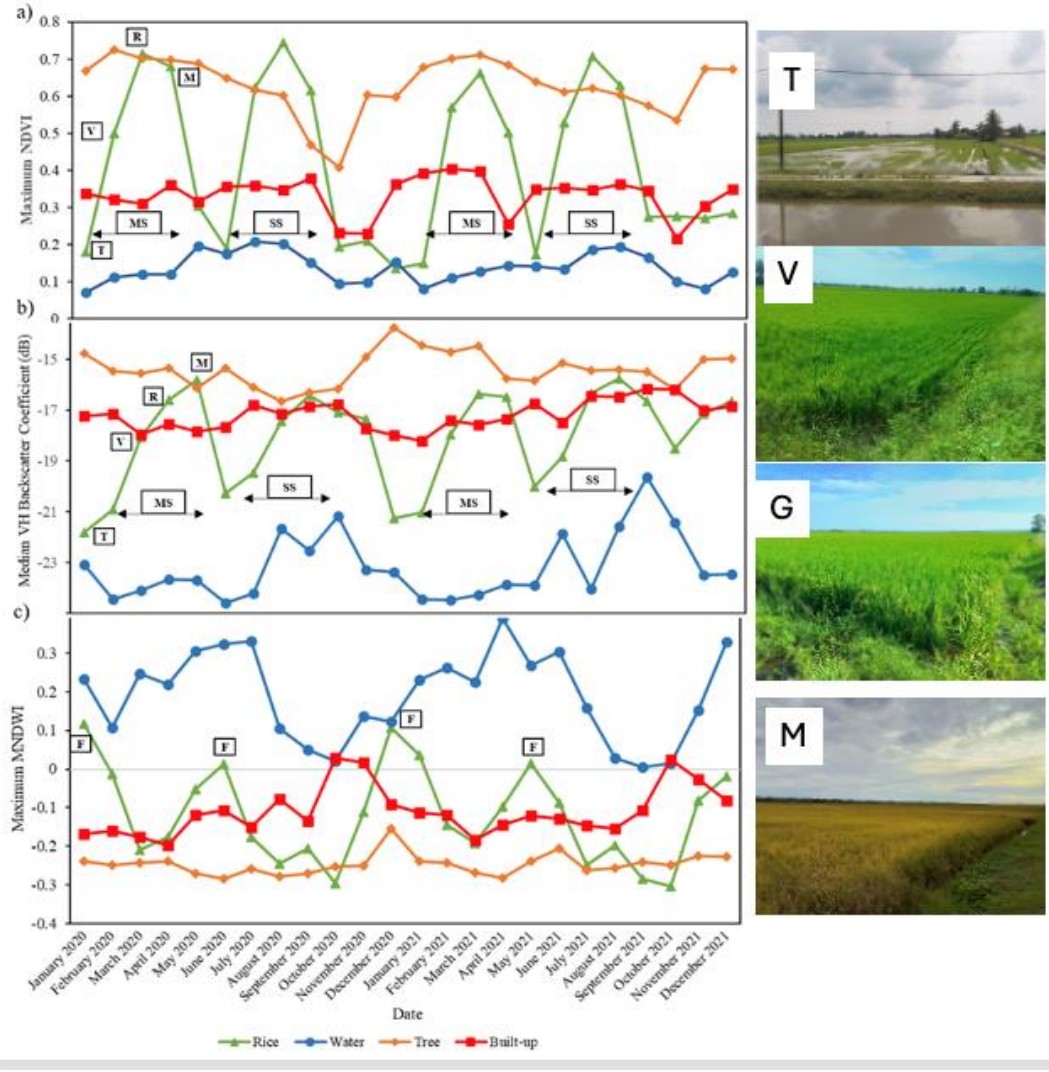

**Figure 3. Temporal signature profiles. (a) NDVI, (b) VH backscatter, and (c) MNDWI of four land cover classes; rice field, water body, tree, and built-up; (d) T = tillage and planting, (e) V = vegetative, (f) G = generative, (g) M = maturity, MS = main season, SS = second season, F = flooding.**

**2.3.6 Accuracy Assessment**

The accuracy of our produced maps was evaluated using two types of assessment: (1) comparison with statistical data and other rice maps at the national, provincial, and regency levels (Table 2), and (2) comparison with existing rice maps by NESEA-Rice10 (Han et al., 2021) and 20mRice-MSEAsia (Sun et al., 2023). We used the coefficient of determination ($R^2$) to measure the correlation between our paddy rice maps, agricultural statistics, and existing products using the following equation. We

used Root mean square error (RMSE) to measure the deviation between identified and statistical areas, which was calculated as:



$$R^2 = \frac{(\sum_{i=1}^{n}(x_i - \bar{x}_i) \, x \, (k_i - \bar{k}_i))^2}{\sum_{i=1}^{n}(x_i - \bar{x}_i)^2 \, x \, \sum_{i=1}^{n}(x_i - \bar{x}_i)^2} \tag{1}$$

$$RMSE = \sqrt{\frac{1}{n} \sum_{t=1}^{n}(IA_t - SA_t)^2} \tag{2}$$

where $n$ is the total number of administrative units, xi represents the mapped paddy rice areas, $\bar{x}_i$ is the corresponding
mean value, $k_{is}$ represents the agricultural statistics or areas from existing rice maps, $\bar{k}_i$ is the corresponding mean value, $SA_t$
and $IA_t$ are the statistical area and identified area of the *i-th* province or district, respectively. The unit of RMSE referred to in
this study is a thousand hectares ($10^3$ ha), and to simplify the description, only the values of RMSE are explained below.

## 3 Results

### 3.1 Spatial distribution of paddy rice in Southeast Asia

Figure 1b shows the spatial distribution of paddy rice growing area, rice harvested area, and rice cropping intensity for nine
main rice-producing nations in Southeast Asia in 2020–2021, with a spatial resolution of 10 m. The data (Table 3 and 4) reveals
a diverse pattern of cultivation practices across different countries. Thailand had the largest paddy field harvested area,
followed by Indonesia, Myanmar, Vietnam, Cambodia, the Philippines, Laos, Malaysia and Timor-Leste. Paddy rice fields
were widely distributed and predominantly found in the valleys and deltas of major regional rivers, including the Mekong
River in Vietnam and the Ayeyarwady River in Myanmar (Schneider and Asch, 2020). We estimated that the total rice growing
area across nine Southeast Asian countries amounted to 28.52 Mha (million ha), which is more than twice the NESEA-Rice
map product (12.90 Mha) (Table 3). The harvested area reached 42.91 Mha, slightly lower than FAOSTAT data (45.08 Mha)
(Table 4).

       Southeast Asia is characterised by a tropical climate, which enables rice cultivation throughout the year. Annual rice
cultivation in Southeast Asia is dominated by single-intensity (51%) and double-intensity (47%), but in some locations with
suitable irrigation, harvesting three times a year is possible (2%). The annual single rice crop was dominant and can be found
in dry regions like Thailand, Myanmar, Laos, Cambodia and Timor-Leste. Annual double rice cropping was common in
Indonesia, Vietnam, Malaysia, and the Philippines. Annual triple rice cropping could only be found in the Mekong Delta in
Vietnam and East Java in Indonesia (Fig. 1b). Further information regarding regional variations in rice is described below. Our
results confirm the report from the Monsoon Rice Calendar and RICA (Mishra et al., 2021; Zhao et al., 2023).




**Table 3.** Growing rice area in this study, other rice area maps, and statistics for nine Southeast Asian countries.

| Country | This Study | | | | Existing map products | | Statistics rice growing area (10³ ha) |
|---|---|---|---|---|---|---|---|
| | Growing area with single season (10³ ha) | Growing area with double season (10³ ha) | Growing area with triple season (10³ ha) | Total rice growing area (10³ ha)* | NESEA-Rice10m growing area (10³ ha) | 20mRice-MSEAsia growing area (10³ ha) | |
| Indonesia | 506.07 | 3,611.44 | 241.03 | 4,358.53 | 3,032.23 | NA | 7,060.27 |
| Timor-Leste | 10.95 | 0.00 | 0.00 | 10.95 | 4.36 | NA | 38.70 |
| Malaysia | 0.57 | 242.45 | 0.00 | 243.01 | 250.89 | NA | 281.62 |
| Philippines | 0.00 | 1,644.57 | 0.00 | 1,644.57 | 1,111.92 | NA | 2,039.55 |
| Vietnam | 152.40 | 2,814.55 | 155.49 | 3,122.44 | 2,295.10 | 3,313.21 | NA |
| Cambodia | 2,375.47 | 774.26 | 0.00 | 3,149.73 | 1,198.39 | 2,819.98 | NA |
| Laos | 735.53 | 51.31 | 0.00 | 786.84 | 161.16 | 853.31 | NA |
| Thailand | 7,084.97 | 2,074.12 | 0.00 | 9,159.09 | 1,453.85 | 12,811.95 | NA |
| Myanmar | 3,671.02 | 2,381.61 | 0.00 | 6,052.64 | 3,398.96 | 5,354.20 | NA |
| **Total** | **14,536.98** | **13,594.30** | **396.51** | **28,527.79** | **12,906.86** | | |

NA : data not available
*Total rice growing area = growing area with single season + growing area with double season + growing area with triple
season

**Table 4.** Harvested rice area in this study and statistics for nine Southeast Asian countries.

| Country | This study | Statistical data | |
|---|---|---|---|
| | Total harvested area (10³ ha)** | National statistics rice harvested area (10³ ha) | FOASTAT harvested area 2021 (10³ ha) |
| Indonesia | 8,452.02 | NA | 10,411.80 |
| Timor-Leste | 10.95 | NA | 26.79 |
| Malaysia | 485.46 | NA | 645.67 |
| Philippines | 3,289.13 | NA | 4,805.08 |
| Vietnam | 6,247.96 | 7,108.90 | 7,219.80 |
| Cambodia | 3,923.99 | 3,264.00 | 3,252.99 |
| Laos | 838.14 | 939.66 | 943.19 |
| Thailand | 11,233.21 | 10,849.58 | 11,244.00 |
| Myanmar | 8,434.25 | 6,909.14 | 6,536.69 |
| **Total** | **42,915.11** | | **45,086.01** |

NA : data not available
**Total harvested area = growing area with single season + (2 x growing area with double season ) + (3 x growing area with
triple season)



### 3.1.1 Indonesia and Timor-Leste

The paddy rice distribution map for Indonesia from 2020 to 2021 reveals a total rice-growing area of 4,358,528 ha and a total harvested area of 8,452,017 ha (Fig. 4a, Table 3 and Table 4). Of this total area, the dominant one was double-intensity with a harvested area of 7,222,870 ha (85.4%), followed by triple-intensity with 723,081 ha (8.6%), then single-intensity with 506,066 ha (6.0%). The five main islands in Indonesia, including Sumatra, Kalimantan, Sulawesi, Java, and Papua, constituted 20% of rice production in Southeast Asia (Supplementary Table S1). However, the densely populated island of Java was the most productive region, mainly in East Java (1,530,864 ha), Central Java (1,308,475 ha), and West Java (1,122,657 ha). The triple rice crop intensity dominant area could be found in East Java, and a few in Central Java, and Lombok (Fig. 4b).

The cropping patterns generated in this study offer detailed information on rice intensity and planting schedule. Figure 4c, d, and e show the representative spectra cluster profiles of the paddy rice NDVI, VH backscatter, and MNDWI for single, double, and triple intensities in East Java, Indonesia. The time series profile of NDVI and VH backscatter can be divided into four periods: T: tillage and transplanting (30 days); V: vegetative (30 days); R: reproductive (30 days); and M: maturity (30 days) (Sianturi et al., 2018; Fatchurrachman et al., 2022) as shown in Fig. 3. Thus, a rice cultivation season lasts three to four months. The lowest NDVI and VH backscatter values for rice during the transplanting phase are 0.12 and -24.89, while the highest values during the generative phase are 0.74 and -15.45, respectively. The peak MNDWI can identify flooding occurring during the tillage and transplanting phases, with values ranging from 0.078 to 0.4. We note that the range of values for this peak may vary depending on locations. Based on this information, the paddy rice cultivation season for single planting is typically from May to September. In wetland areas, paddy rice was planted during the transition between the rainy and dry seasons. Areas with double cropping typically had their first season from January to May during the rainy season and the second from June to September during the transition between the rainy and dry seasons. In regions with triple cropping intensity, the first season spanned from March to July, the second from July to November, and the third from November to March. During the third season, commonly in the dry season, farmers used groundwater for irrigation through pumps. Using NDVI and VH backscatter phenology, the mean duration of a rice growing season (for single, double, and triple) was calculated to be about 120 days.

The spatial distribution map and cropping intensity of paddy rice in Timor-Leste during the growing season of 2020 to 2021 are depicted in Fig. 5a. Timor-Leste has a comparatively smaller area of rice fields compared to other Southeast Asian countries. The rice fields in Timor-Leste were distributed mostly in the riverbank and river delta along the north and south coastlines, covering 10,592 ha of harvested area (Table 3). In Timor-Leste, rice fields were mostly planted once a year. This pattern is attributed to the relatively low rainfall in the country. Lowlands rice fields were dominant in regions along the southern coast, including Viqueque, Covalima, and parts of Lautém Municipality, as well as the north coast encompassing Lautém, Manatuto, and much of Bobonaro Municipality (Fig. 5b). Paddy fields in this region were irrigated (Kumashiro Teruyoshi, 2013).



(a) Indonesia

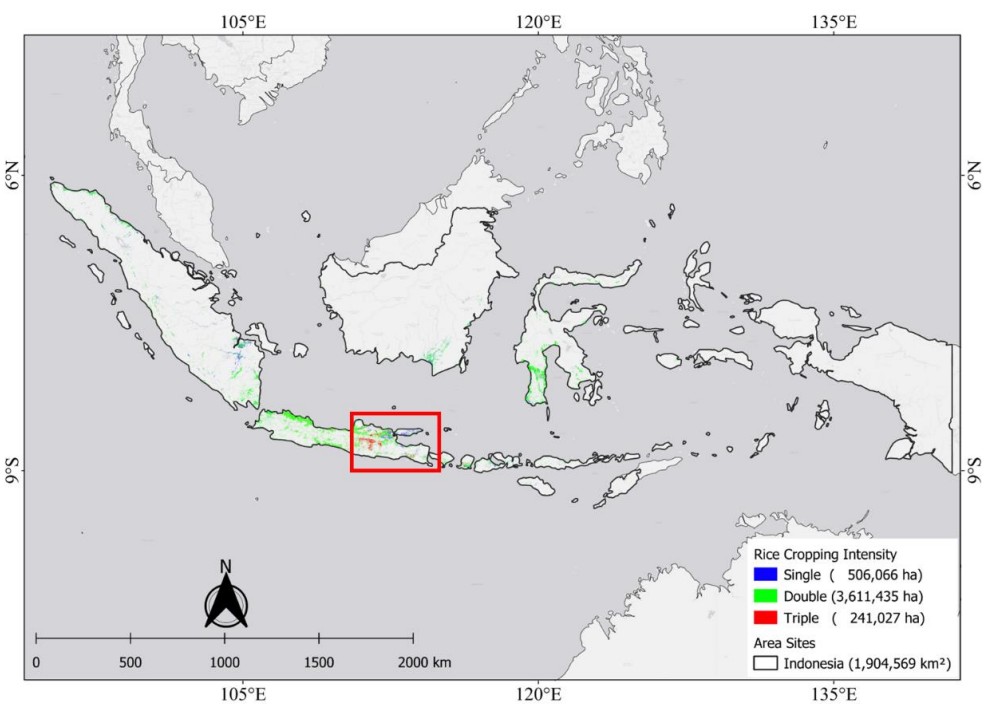

(b) Java Island

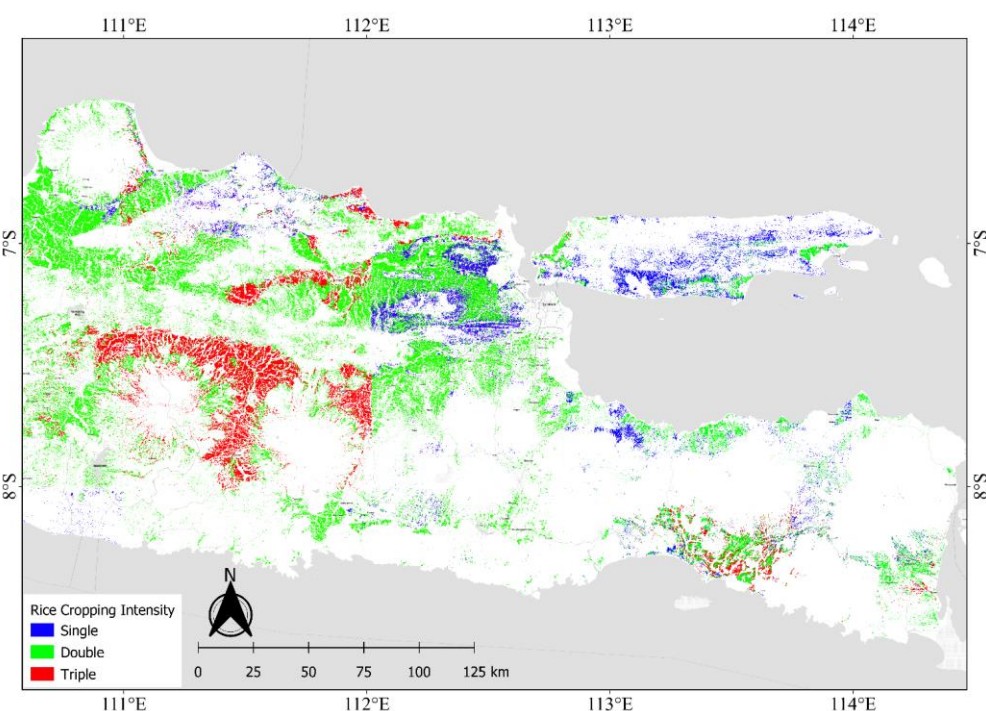



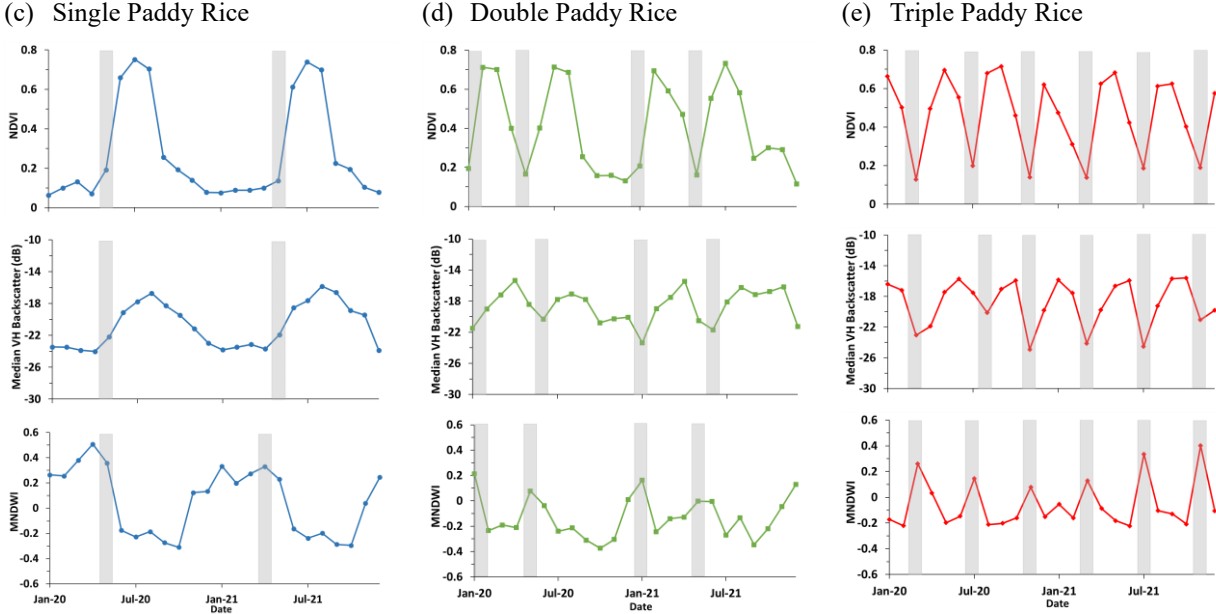

**Figure 4. Map and cropping intensity of paddy rice in (a) Indonesia, (b) East Java, and representative spectra cluster profiles of the paddy rice NDVI, VH backscatter, and MNDWI in the East Java province, Indonesia: (c) Single paddy rice, (d) Double paddy rice, and (e) Triple paddy rice. Grey bars refer to the initial season (i.e., transplanting period).**

The representative spectra cluster profiles of the paddy rice NDVI, VH backscatter, and MNDWI patterns for single

rice cultivation in the North Coast of Timor-Leste are illustrated in Fig. 5c. The paddy rice planting season occurred between

April to August during rainy season. High precipitation is observed in northern and southern areas from January to April. The

average annual precipitation in Timor-Leste was less than 2000 mm, with a monthly average ranging from 30 to 370 mm

(Kumashiro Teruyoshi, 2013). During the transplanting phase, rice areas displayed lowest NDVI and VH values at 0.15 and -

24.87, respectively. Conversely, in the generative phase, the highest values observed were 0.71 for NDVI and -16.15 for VH.

The high MNDWI pattern identified flooding during the tillage and transplanting phases, with values ranging from 0.11 to

0.16. The average duration of a single season of paddy rice, as determined by NDVI and VH phenology, ranged from 120 to

150 days.






**Figure 5. Map and cropping intensity of paddy rice in (a) Timor-Leste, (b) North Coast of Timor-Leste, and (c) representative spectra cluster profiles of the paddy rice NDVI, VH backscatter, and MNDWI in the North Coast of Timor-Leste. Grey bars refer to the initial season (i.e., transplanting period).**



### 3.1.2 Malaysia

Figure 6a displays the spatial distribution map and cropping intensity of paddy rice in Malaysia during the growing season from 2020 to 2021. Malaysia had the second smallest rice field area among Southeast Asian nations. The total area of rice fields was approximately 243,014 ha (Table 3), and the harvested area was doubled at 485,460 ha (Table 4). Double intensity paddy fields dominated across Peninsular Malaysia (harvested area of 484,896 ha or 99.8%, especially in the granary areas equipped with irrigation infrastructure, and only a few areas with a single crop (566 ha or 0.2%) in East Malaysia/Borneo

States (Sabah). Figure 6b shows that the northern region of Peninsular Malaysia, including the states of Kedah, Perak, and Perlis, the largest rice field area with double cropping (Fatchurrachman et al., 2022).

Figure 6c and 6d present the representative spectra cluster profiles of the paddy rice NDVI, VH backscatter, and MNDWI patterns for single and double rice cultivation intensities in Malaysia. Peninsular Malaysia had double-intensity rice cultivation: the first season from January to May or June, and the second from July to November or December. Meanwhile,

Sarawak and Sabah States at Borneo had mainly single-intensity paddy rice planting seasons, with the single planting typically taking place from October to March. Peninsular Malaysia had two cropping seasons, the main off-season, with no cash crops in between seasons (Fatchurrachman et al., 2022). Using NDVI and VH phenology, the mean duration of a season was calculated to be about 120 days. During the transplanting phase, rice areas exhibited the lowest NDVI and VH values at 0.13 and -25.13, respectively. Conversely, in the generative phase, the highest values observed were 0.73 for NDVI and -16.48 for

VH backscatter. The high MNDWI pattern identified flooding during the tillage and transplanting phases, with values spanning from -0.05 to 0.15.








(a)  Malaysia

(b)  Kedah and Perlis States of Malaysia

(c)  Single Paddy Rice

(d)  Double Paddy Rice

**Figure 6. Map and cropping intensity of paddy rice in (a) Malaysia, (b) Kedah and Perlis States of Malaysia, and representative spectra cluster profiles of the paddy rice NDVI, VH backscatter, and MNDWI in Kedah and Perlis, Malaysia: (c) Single paddy rice, and (d) Double paddy rice. Grey bars refer to the initial season (i.e., transplanting period).**




### 3.1.3 The Philippines

Figure 7a shows the spatial distribution map of paddy fields and cropping intensity in the Philippines during the cropping season from 2020 to 2021. The Philippines had the sixth-largest area of rice fields among Southeast Asian nations. Rice fields
were all planted twice annually with a total harvested area of 3,289,130 ha (Table 4). Double rice cropping are mostly practiced in the Philippines, as reported Laborte et al., 2012. Largest rice fields area could be found in Nueva Ecija, Isabela, and Cigayan provinces in the Luzon Islands (Fig. 7b).

The representative spectra cluster profiles of the paddy rice NDVI, VH backscatter, and MNDWI patterns in the Northern Luzon of Philippines is presented in Fig. 7c. The figures show the typical characteristics with the first season from
July to October/November and the second season from January to May. The average duration of a season, as determined by NDVI and VH phenology, ranged from 120 to 150 days. During the transplanting phase, the lowest recorded values for NDVI and VH were 0.28 and -22.71, respectively. In contrast, the highest values during the generative phase were 0.76 for NDVI and -16.26 for VH. The peak MNDWI profile demonstrates the effectiveness in detecting flooding during tillage and transplanting phases, with values ranging from -0.09 to 0.07.

### 3.1.4 Vietnam

Figure 8a shows the spatial distribution map and cultivation intensity of paddy rice in Vietnam during the 2020 to 2021 growing seasons. The total rice growing area was 3,122,438 ha (Table 3), with a total rice harvested area that was more than double at 6,247,962 ha (Table 4). This study indicates that Vietnam had the fifth-largest expanse of rice fields within the Southeast Asian region. The total harvested area was dominated by double-intensity paddy rice with an area of 5,629,108 ha (90%), followed
by triple-intensity with 466,455 ha (8%) and single-intensity with 152,399 ha (2%). The rice-producing central areas in Vietnam were concentrated in the Mekong River Delta (Kein Giang, An Giang, and Long An provinces) and the Red River Delta (Nam Dinh and Thai Binh provinces). The Mekong River Delta was the primary location of the triple rice crop intensity (Fig. 8b) (Han et al., 2022).

The representative spectra cluster profiles of the paddy rice NDVI, VH backscatter, and MNDWI patterns for single,
double, and triple rice cultivation intensities in Mekong Delta, Vietnam are shown in Fig. 8c, d, and e. The paddy rice cultivation for the single season typically spanned from October to February. In the case of double-intensity cropping, the first season occurred from May/June to September/October, while the second was from November to March/April. The triple intensity involved: the first season from April to August, the second from August to December, and the third from December to April. The typical length of a season, as identified through the analysis of NDVI and VH phenology, varied from 90 to 120
days. The triple-intensity paddy rice fields in the Mekong River Delta have advanced dike and canal systems (Guan et al., 2016). The minimum NDVI and VH values for rice in the transplanting phase were 0.08 and -23.69, respectively, whereas the maximum values in the generative phase were 0.79 for NDVI and -15.03 for VH. The peak MNDWI profile effectively detects flooding during the tillage and transplanting phases, with values ranging from -0.11 to 0.33 in Vietnam.





**Figure 7. Map and cropping intensity of paddy rice in (a) Philippines, (b) Northern Luzon of Philippines, and representative spectra**
**cluster profiles of the paddy rice NDVI, VH backscatter, and MNDWI in the Northern Luzon of Philippines: (c) Double paddy rice.**
**Grey bars refer to the initial season (i.e., transplanting period).**

**Figure 8. Maps and cropping intensity of paddy rice in (a) Vietnam, (b) Mekong Delta in Vietnam, and representative spectra cluster profiles of the paddy rice NDVI, VH backscatter, and MNDWI in Mekong Delta, Vietnam: (c) Single paddy rice, (d) Double paddy rice, and (e) Triple paddy rice. Grey bars refer to the initial season (i.e., transplanting period).**





### 3.1.5 Cambodia and Laos

The spatial distribution map and cultivation intensity of rice fields in Cambodia for the year 2020 to 2021 are shown in Fig. 9a. The rice field in Cambodia was around 3,149,731 ha (Table 3), with a harvested area of 3,923,990 ha (Table 4). The dominant rice cultivation in Cambodia was a single intensity, with a total harvested area of 2,375,473 ha (61%) and double intensity covering 1,548,516 ha (39%). Rice fields in Cambodia were concentrated in the lowlands surrounding Lake Tonle Sap and the lower reaches of the Mekong River in the southern part of the country. The areas of Cambodia where paddy fields were primarily found in the northwestern (Siem Reap, Banteay Meanchey, and Battambang provinces), central (Kampong Cham), and southern regions (Prey Veng). The paddy planting areas were small and dispersed in the east and northeast, which is strongly related to the topography and water resources in the study area (Kang et al., 2022). Most paddy fields in Fig. 9b were near Tonle Sap Lake and the Mekong River in low-altitude areas.

Figures 9c and 9d display the representative spectra cluster profiles of the paddy rice NDVI, VH backscatter, and MNDWI patterns for single and double rice cropping intensities in the Northwestern region of Cambodia. Single paddy rice cultivation typically commenced in May/June and concluded in November/December. The double-intensity season included the first season from May/June to October/November and the second from October to May. During the transplanting stage, the NDVI and VH values for paddy rice were estimated to be 0.01 and -28.42. Conversely, the values reached their highest point during the generative stage, with NDVI peaking at 0.69 and VH reaching -14.57. The harvesting period for rice at a single planting intensity, as observed through spectral NDVI, was relatively extended, lasting approximately 5 to 6 months. The annual Cambodia rainfall regime includes a five-month dry season (December to April) and a longer wet season (May to November) (Chhinh and Millington, 2015). The MNDWI values peaked from -0.17 to 0.48 during the stages of tillage and transplanting.





Figure 9. Map and cropping intensity of paddy rice in (a) Cambodia, (b) Northwestern region of Cambodia, and representative spectra cluster profiles of the paddy rice NDVI, VH backscatter, and MNDWI in the Northwestern region of Cambodia: (c) Single paddy rice, and (d) Double paddy rice. Grey bars refer to the initial season (i.e., transplanting period).






Figure 10a presents the spatial distribution and cultivation intensity of paddy rice in Laos during 2020 and 2021, with a rice growing area of 786,835 ha (Table 3) and a harvested area of 838,144 ha (Table 4). The main crop in Laos was paddy rice, which covers a harvested area of 735,526 ha (87.8%) for single-intensity cultivation and 102,618 ha (12.2%) for double-intensity cultivation. Paddy rice areas were primarily spread across the Central regions of Laos (Savannakhet and Khammouan

provinces) and Southern regions of Laos (Champasak and Salavan provinces). The paddy planting areas were small and dispersed in the north, which is strongly related to the landscape and supply of water in the study area (Xiao et al., 2006). As shown in Fig. 10b, most paddy fields in central Laos were located on hillside terrain and adjacent to streams.

      The spectra cluster profiles of the paddy rice NDVI, VH backscatter, and MNDWI patterns for single and double rice cultivation intensities per year in the central region of Laos are shown in Fig. 10c and d. Single planting typically from

June/July to November/December, and double intensity consisted of the first season (July to December/January) and the second season (January to June/July). The peak monthly rainfall was observed between July and September in Laos (Tsubo et al., 2006). Total rainfall in the Northern, Central, and Southern regions range from 1284 to 1580 mm/year, 1131 to 1791 mm/year, and 2105 to 3359 mm/year, respectively (Basnayake et al., 2006). In the transplanting stage, the estimated NDVI and VH values for paddy rice were 0.25 and -22.55, respectively. On the other hand, the values reach their maximum during the

generative stage of rice, with NDVI peaking at 0.67 and VH reaching -13.92. The MNDWI values reached their highest points from 0.02 to 0.11 during the tillage and transplanting stages.








Figure 10. Map and cropping intensity of paddy rice in (a) Laos, (b) Central region of Laos, and representative spectra cluster profiles of the paddy rice NDVI, VH backscatter, and MNDWI in the Central region of Laos: (c) Single paddy rice, and (d) Double paddy rice. Grey bars refer to the initial season (i.e., transplanting period).





### 3.1.6 Thailand and Myanmar

Figure 11a show the paddy rice distribution map in Thailand from 2020 to 2021, with an estimated 9,159,089 ha of rice fields
and an annual harvested area of 11,233,205 ha (Tabel 3). This made Thailand the largest rice producer in Southeast Asia. The
dominant rice cropping in Thailand was single-intensity with an area of 7,084,973 ha (i.e., 63.2% of total harvested area) and
double-intensity with 4,148,232 ha (36.8%). The largest paddy rice cultivation area in Thailand is the region of Northeast
Thailand, like Ubon Ratchathani, Surin, Buriram, Nakhon Ratchasima, Si Saket, and Roi Et, all have distinct dry and wet
seasons. Rain-fed single-season rice is the primary crop in this region due to its primary water source problem (Xu et al., 2021),
where its mean annual rainfall is only about 1400 mm (Gale and Saunders, 2013). Figure 11b demonstrates an extensive total
paddy rice cultivation area in Northeast Thailand, but individual plots were small and irregular. Meanwhile, the central plain
had a higher urbanization rate and advanced irrigation infrastructure, offering a more suitable location for a rice double
cropping system. In northern regions such as Chiang Rai and Chiang Mai, rice fields were concentrated in the valleys. The
distribution of paddy rice in Southern Thailand was relatively small compared to other regions. Nevertheless, the map produced
in this study can effectively extract small, discrete rice paddy plots (e.g. Fig. 17).

Figures 11c and 11d show the representative spectra cluster profiles of the paddy rice NDVI, VH backscatter, and
MNDWI for single and double rice cropping intensities in the Northeast region of Thailand. In Thailand, paddy rice was grown
in two cropping periods, including the wet and dry seasons. Paddy rice cultivation season for single planting was mainly sown
from May/June and harvested in November/December (Fig. 11c). This single-planting cultivation period is commonly referred
to as off-season rice. Off-season rice can be planted at various times of the year outside the wet and dry seasons, depending on
the region and water availability (Sanwong et al., 2023). The age at harvest for off-season rice can vary, but it is often around
4 to 6 months. Meanwhile, rice with double intensity is sown both during the wet season (May to July) and dry season (August
to December/January) (Fig. 11d). The NDVI and VH values for paddy rice during the transplanting stage were approximately
0.21 and -23.67. In contrast, the values peak during the generative stage of rice, reaching 0.71 for NDVI and 14.53 for VH.
MNDWI values peaked between -0.08 and -0.02 during tillage and transplanting.





Figure 11. Map and cropping intensity of paddy rice in (a) Thailand, (b) Northeast region of Thailand, and representative spectra cluster profiles of the paddy rice NDVI, VH, and MNDWI in the Northeast region of Thailand: (c) Single paddy rice, and (d) Double paddy rice. Grey bars refer to the initial season (i.e., transplanting period).



The spatial distribution map and cultivation intensity of paddy fields in Myanmar for the 2020 to 2021 cropping
seasons are shown in Fig. 12a. The rice field parcel in Myanmar covered an approximate area of 6,052,636 ha (Table 3), with
a harvested area of 8,434,250 ha (Table 4). The rice-harvested area within single intensity is 3,671,024 ha (43.5%), while
4,763,224 ha (56.5%) on double intensity. Paddy rice with single intensity was dominated in the North of Myanmar,
particularly in the central highland regions (i.e. Mandalay, Sagaing, and Magway) and few with double intensity. In general,
single-cropped rice was invariably practised during the rainy season, and irrigation was a major constraint for double-cropped
rice production during the dry season in the central highland areas (Son et al., 2017). In contrast, the double intensity of paddy
rice was more widely distributed throughout coastal regions (such as Ayeyarwady, Yangon, and Bago) (Fig. 12b). The double-
intensity paddy rice was concentrated along the main rivers, especially in the Ayeyarwady river basin, due to water availability
for crop irrigation in both cropping seasons (Son et al., 2017).

Figures 12c and d depict the representative spectra cluster profiles of the paddy rice NDVI, VH backscatter, and
MNDWI patterns for single and double rice cultivation intensities in the Mandalay region of Myanmar. The single planting
was typically from July/August to December and double intensity consisted of the first season (July/August to December) and
the second (May to July). Research by (Torbick et al., 2017) shows a high correlation between the wet season rice calendar
dates for March and November and the dry season from November to March. During the transplanting stage, the estimated
NDVI and VH values for paddy rice were 0.15 and -20.45, respectively. In contrast, the generative stage of rice has its peak
values, with NDVI reaching a maximum of 0.73 and VH reaching -14.85. The MNDWI values peaked from -0.04 to 0.07
during the tillage and transplanting stages.






**Figure 12. Map and cropping intensity of paddy rice in (a) Myanmar, (b) Mandalay region of Myanmar, and representative spectra cluster profiles of the paddy rice NDVI, VH backscatter, and MNDWI in the Mandalay region of Myanmar: (c) Single paddy rice, and (d) Double paddy rice. Grey bars refer to the initial season (i.e., transplanting period).**



## 3.2 Comparison with national agricultural statistics

We compared our results with agricultural statistics for the nine countries of Southeast Asia at the district and provincial level (Table 2). The agricultural statistical data utilised in this study includes Indonesia, the Philippines, and Malaysia (Fig. 13). In contrast, the annual total harvested area is available for Thailand, Cambodia, Laos, Myanmar, and Vietnam (Fig. 14). The results indicate a strong agreement between our estimated rice areas and the statistical data.

Figure 13 illustrates the comparison results between our results and agricultural statistics concerning the paddy field parcels at both the district and provincial levels in Indonesia, Malaysia, and the Philippines. The distribution of paddy field parcels in classification and agricultural statistics showed a relatively high correlation in Indonesia ($R^2$ = 0.85) (Fig. 13a). Figure 14a also shows that our predicted data of Indonesia were consistently lower than the statistical data, with a mean error of 6.46 x $10^3$ ha and RMSE = 11.44 × $10^3$ ha. The reason may be that statistical data from Indonesia include rice areas intermingled with maize, sugar cane, tobacco, and other seasonal crops, leading to the overestimation of these crops as rice fields. Our estimated paddy field growing areas in Malaysia and the Philippines were strongly correlated with district and provincial agricultural statistics, with $R^2$ value of 0.97 (Fig. 13b and 13c).

A comparison between the harvested area in Southeast Asia, derived from this study and FAOSTAT 2021 at the national level is presented in Table 4. The total harvested area from this study is 42,915,110 ha, which closely corresponds to FAOSTAT's data (45,086,008 ha harvested area). The relative discrepancy is approximately 4.3% with $R^2$ value of 0.92 and RMSE value of 1,114 x $10^3$ ha (Supplementary Fig. S1).

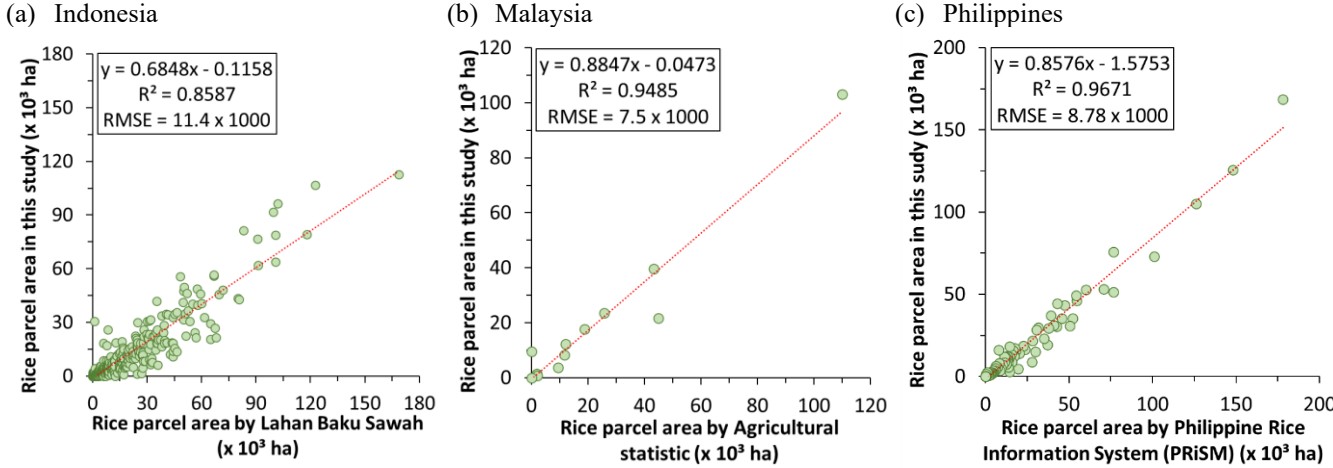

**Figure 13. The comparison of paddy rice growing areas from our map and the statistical data at district and province levels for (a) Indonesia (2021-2022), (b) Malaysia (2021), and (c) Philippines (2022).**



Figure 14 compares our estimate with agricultural statistics for the total annual harvested rice area at the provincial level in Vietnam, Cambodia, Laos, Thailand, and Myanmar. Overall, our estimates were consistent with the statistical data of

the countries. The results show a high correlation in Cambodia ($R^2 = 0.9$) (Fig. 14b), with RMSE = 63.45 x $10^3$ ha. The estimated harvested areas for paddy fields in Myanmar, Thailand, Laos, and Vietnam were also strongly correlated with agricultural statistics ($R^2$ values ranging from 0.93 to 0.97) (Fig. 14a, c, d, and e).

**Figure 14. The comparison of harvested paddy rice areas from our map and statistical data at the province levels for (a) Vietnam (statistical data from 2021), (b) Cambodia (2019), (c) Laos (2021), (d) Thailand (2021), and (e) Myanmar (2020).**


### 3.3 Comparison with existing rice area maps

We further compared the rice maps obtained in this study with existing maps available at the district levels. The datasets include the combined MODIS and Sentinel-1 rice paddy map with a resolution of 10 m in Northeast and Southeast Asia from 2017 to 2019 (NESEA-Rice10 product). Additionally, there was a Sentinel-1 rice paddy map with a resolution of 20 m





specifically for mainland Southeast Asia in 2019 (20mRice-MSEAsia) (see Section 2.2.3 for details). Figure 15 demonstrates the correlation between the paddy rice area obtained from our maps and existing products in Southeast Asia. The correlation coefficients ranged from $R^2=0.28$ (for Thailand) to $R^2=0.98$ (for Malaysia), indicating varying degrees of correlation between all datasets.

Figures 15a, f, g, and h show the correlations between the paddy rice area from our estimates and the NESEA-Rice10

products for Indonesia, Cambodia, Laos, and Thailand with $R^2$ values of 0.63, 0.50, 0.59, and 0.28. The NESEA-Rice 10 product estimated that the rice acreage across Southeast Asia is underestimated. The NESEA-Rice10 m products exhibit the lowest significant underestimation in the rice-growing areas, particularly in Thailand (Northeastern region), Laos (Southern region), and Indonesia (Java Island) (Fig. 16a, c, d, f, g and h). The total growing paddy rice area from NESEA-Rice10 m data accounts for only 16% (Thailand), 20% (Laos), and 70% (Indonesia) compared to our results.

While our map demonstrates a strong correlation in paddy rice area with NESEA-Rice10, with $R^2$ values of 0.86, 0.98, 0.84, 0.94, and 0.86 in Timor-Leste, Malaysia, Philippines, Vietnam, and Myanmar (as illustrated in Fig. 15b, c, d, e, and i), it is worth noting that our results were notably (55%) larger compared to those obtained from NESEA-Rice10 (Supplementary material Table S3).

The correlation between the paddy rice parcel area reported in agricultural statistics with the NESEA-Rice10 product

is $R^2$ values of 0.77, 0.96, and 0.86, respectively, and is only available for Indonesia, Malaysia, and the Philippines (Table 5). No statistical data is available on the paddy rice growing area in the other five nations.

**Table 5.** The paddy rice parcel and harvested area correlations between this study, and NESEA-Rice10 with agricultural statistics

| Country | This study | | NESEA-Rice10 | |
|---|---|---|---|---|
| | RMSE (x $10^3$ ha) | $R^2$ | RMSE (x $10^3$ ha) | $R^2$ |
| Indonesia | 11.4[a] | 0.85[a] | 236.3[a] | 0.77[a] |
| Timor-Leste | 0.18[a] | 0.86[a] | NA | NA |
| Malaysia | 7.5[a] | 0.94[a] | 7.34[a] | 0.96[a] |
| | 25.7[b] | 0.84[b] | NA | NA |
| Philippines | 8.7[a] | 0.96[a] | 18.4[a] | 0.86[a] |
| Vietnam | 37.3[b] | 0.96[b] | NA | NA |
| Cambodia | 63.4[b] | 0.90[b] | NA | NA |
| Laos | 11.8[b] | 0.98[b] | NA | NA |
| Thailand | 27.2[b] | 0.97[b] | NA | NA |
| Myanmar | 240.9[b] | 0.93[b] | NA | NA |

NA : The paddy rice growing or parcel area in agricultural statistical data is not available.

[a] : Paddy rice growing or parcel area

[b] : Paddy rice harvested area







(a) Indonesia  (b) Timor-Leste  (c) Malaysia
(d) Philippines  (e) Vietnam  (f) Cambodia
(g) Laos  (h) Thailand  (i) Myanmar

**Figure 15. Comparisons of the paddy rice parcels area between our study and existing datasets at the district level in (a) Indonesia, (b) Timor-Leste, (c) Malaysia, (d) Philippines, (e) Vietnam, (f) Cambodia, (g) Laos, (h) Thailand, and (i) Myanmar.**




Figures 15e, f, g, h, and I illustrate a high correlation between the paddy rice area from our map and 20mRice-MSEAsia for Vietnam, Cambodia, Laos, Thailand, and Myanmar with $R^2$ values of 0.91, 0.76, 0.78, 0.73, and 0.79, respectively. The 20mRice-MSEAsia overestimates paddy area by approximately 40% (Thailand) and 8% (Laos) compared to our study. The discrepancy is notable, particularly in the Central and Northeast regions of Thailand and South of Laos (Fig. 16 a, b, d, and e).

| (a) This study (Northeast of Thailand) | (b) 20mRice-MSEAsia (Northeast of Thailand) | (c) NESEA-Rice10 (Northeast of Thailand) |
|---|---|---|

| (d) This study (Southern of Laos) | (e) 20mRice-MSEAsia (Southern of Laos) | (f) NESEA-Rice10 (Southern of Laos) |
|---|---|---|

| (g) This study (Java Island) | No available data | (h) NESEA-Rice10 (Java Island) |
|---|---|---|

**Figure 16. Visual comparison between our paddy rice maps and existing products in typical regions: (a, d, and g) classification our study with the location within each country (inset); (b and e) 20mRice-MSEAsia product; (c, f, and h) NESEA-Rice10 product.**



## 4 Discussion

### 4.1 Advantages of the local unsupervised classification with phenological labelling (LUCK-PALM) method

This study optimised a local unsupervised and penological mapping method with a combination of Sentinel-1 and Sentinel-2 time series data to capture paddy rice distribution and rice cropping intensity, namely the LUCK-PALM method. The phenology-based classification method for mapping rice cropping intensity requires dense time series imagery data. MODIS imagery data with excellent temporal resolution (i.e., 1 to 2 days if clouds do not impact observations) have been used for mapping of rice cropping intensity for a larger extent. However, the coarse spatial resolution leads to high uncertainty (Laborte et al., 2017; Mishra et al., 2021; Zhao et al., 2023). In this study, we successfully produced a more detailed map across Southeast Asia (Fig. 1b) by using a combination of Sentinel-1 and 2 time series data. Although Sentinel-2A and 2B have a temporal resolution of 5 days, we used monthly composite image data of NDVI and MNDWI to reduce noise and the effect of cloud cover. These data can capture rice phenological growth stages, including transplanting, vegetative, generative, and ripening stages, excellently, as each stage lasts about 30 days (Rudiyanto et al., 2019; Fatchurrachman et al., 2022).

The median of the VH polarisation can distinguish more pronounced values of VH backscatter at various rice growth stages (Minh et al., 2019; Rudiyanto et al., 2019). The enhanced temporal frequency of Sentinel-1 images significantly improved the precision of rice growth stage determination, as it allowed for more accurate backscattering profile assessments. Nevertheless, the VH backscatter in arid regions could not detect water effectively. Therefore, integrating Sentinel-1 and Sentinel-2 data can mitigate the limitations of a single data source, improving the overall results of mapping paddy rice. SAR data may penetrate cloud cover and supplement optical data in continuous Earth surface monitoring to correct for missed phenological phenomena. The accuracy of rice identification with integrated Sentinel-1 and Sentinel-2 images is greater than with Sentinel-1 or Sentinel-2 images alone (Jiang et al., 2023).

Advanced supervised classification methods always require substantial training data, which is a challenge. Southeast Asia has varying rice cropping intensity and calendar, and collecting this data can be tedious and time-consuming. In this study, we employed a local unsupervised and phenological mapping method. This allowed us to generate representative cluster profiles of paddy NDVI, VH backscatter, and MNDWI, which can efficiently identify varying rice cropping intensity with different cropping calendars. Thus, the proposed LUCK-PALM method stands as a cost-effective approach for mapping rice cropping intensity on a large area with varying climate conditions.

In this study, we not only mapped the distribution of rice growing area but also cropping intensity at the pixel level. This information can be used to predict rice yields (Pan et al., 2021; Cao et al., 2021), estimate $CH_4$ emissions (Sun et al., 2020; Zhang et al., 2020a), optimise water use (Liu et al., 2021; Li et al., 2022), and assess the effects of climate change (Mahajan et al., 2012; Ye et al., 2015).





## 4.2 Comparison of rice growing area with other rice paddy map products

The map produced in this study shows high agreement with countries' agricultural statistics data (Table 5) and was verified manually using Google Images' Street view (Results in Supplementary Fig. S2 and Table S2). Compared with existing rice maps, our study presents higher correlations with ground data (Table 5).

Sentinel-1 data is widely used to map rice extents, especially in tropical regions because the C-band SAR sensor can obtain data through cloud cover (Clauss et al., 2018; Rudiyanto et al., 2019; Singha et al., 2019; Vuong et al., 2019; Phan et al., 2021; Xu et al., 2021; Phung et al., 2022; Sun et al., 2023). In an earlier study, the 20mRice-MSEAsia using Sentinel-1 data calculated the total area of rice fields in mainland Southeast Asia is 25,150 x $10^3$ ha (Sun et al., 2023). This value is greater than the rice growing area estimated in this study (22,496 x $10^3$ ha). In contrast, the integration of the phenological rule-based

method with MODIS and Sentinel-1 data by (Han et al., 2021) or NESEA-Rice10 product indicates that the rice growing areas in Southeast Asia were 12,902 x $10^3$ ha, which was considerably lower compared to our product. The significant difference between our product and those two existing products is particularly notable in Thailand (Fig. 16a, b, c, Table 3 and Supplementary Table S3). As shown in Fig. 17a, our method can identify the distribution of small rice fields (average 0.2 ha) located in narrow valleys (i.e., streamlines) in mountainous areas of the dry region in Northeast Thailand. Both existing

products failed to detect these fields. The 20mRice-MSEAsia product considered non-rice as rice areas (Fig. 17b), which indicates it has a high commission error of rice groups, while NESEA-Rice10 classified rice as non-rice areas (Fig 17c), which suggests it results in a high omission error of rice groups. The overestimation of 20mRice-MSEAsia product (especially for Thailand, Table 3) suggests that the use of Sentinel-1 data alone with the semantic segmentation model could not effectively discriminate between rice and other crops, such as sugar cane, maize, and cassava (Fig. 17b, d, and e). In contrast, the

underestimation of NESEA-Rice10 product (Table 3) indicates the phenological rule-based method with MODIS and Sentinel-1 data of (Han et al., 2021) could not effectively identify rice growing areas, especially in the regions with relatively low rainfall intensity like the mainland Southeast Asia. The main factor contributing to this discrepancy was that the regions lack strong flooding signals that can be detected by the radar (Zhang et al., 2020a). In addition, the coarser MODIS data missed many small rice fields located in narrow valleys between the mountains (Fig. 17c). Most paddy rice is grown annually as a

rainfed system due to low precipitation and lack of irrigation water in Northeast Thailand (Arunrat et al., 2020), Cambodia (Fukai and Ouk, 2012), and Laos (Xiao et al., 2006). These comparisons (Fig. 17a, b, and c), suggest that our results are more accurate compared to the two existing products.

        Other studies using MODIS data at a coarser resolution of 250 and 500 m suggested a total rice growing area in Southeast Asia of 33,939 x $10^3$ ha (Xiao et al., 2006) and 42,335 x $10^3$ ha (Bridhikitti and Overcamp, 2012). These estimates

are twice as extensive as the data estimated from our study. These results indicate that using low-resolution MODIS data tends to lead to an overestimation of rice areas, particularly in the border regions where rice and non-rice fields intermingle, creating mixed pixels.


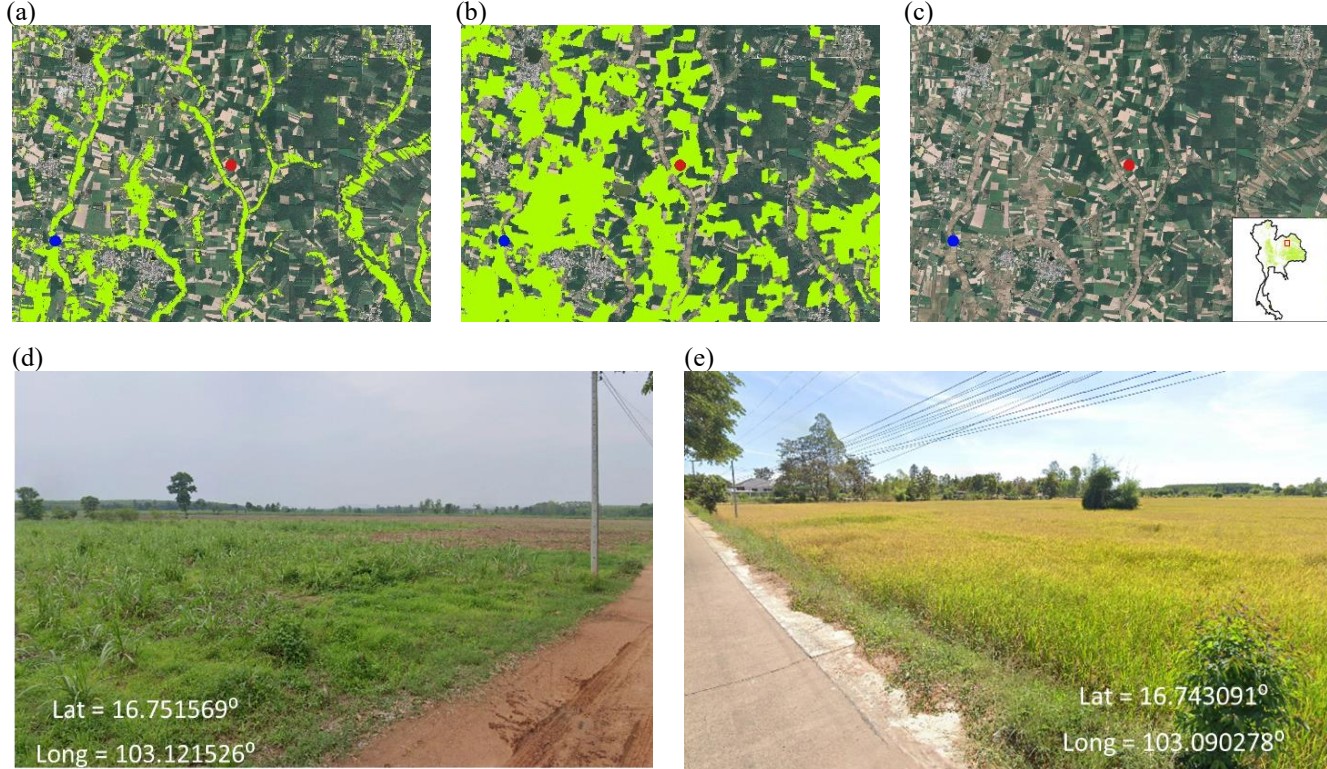

**Figure 17. Visual comparison between rice area maps in the Northeast Thailand superimposed on © Google Earth. (a) Results from this study. (b) 20mRice-MSEAsia product. (c) NESEA-Rice10 product, which does not identify rice fields in the same area. (d) Image from © Street View imagery, which indicates the red point (non-rice area), and (e) which indicates the blue point (rice field).**

**4.3 Uncertainty of our results**

While the outcomes of this study's paddy rice map demonstrate great accuracy, it is essential to acknowledge certain limitations. There are errors in mixed pixels at the intersection of rice and other landscape classes that could be easily overlooked or misclassified (Fig. 18a, b, c, d, e, and f). For example, wetlands with non-rice crops could be mistaken for rice, especially single-cropping rice. Mixed pixels are also present in non-rice fields (such as small roads, levees, and bunds) near

rice fields classified as rice fields. Other errors can be found in rice farming in mountainous and hilly terrains, which are typically small-scale, dispersed, and, in most cases, contain multiple land types mixed with agricultural vegetation (horticulture), creating mixed pixels. Moreover, thick cloud layers, frequent rains, and foggy weather in these areas make it difficult to map rice field areas accurately. The occurrence of mixed pixels is also attributed to the conversion of paddy rice from rice fields to horticultural crops, resulting in water detection during the rainy season on the previously designated rice

plantations. In rice mapping, the mixed pixel effects have previously been reported and have become a source of spatial



uncertainty (Xiao et al., 2006; Clauss et al., 2016; Singha et al., 2019; Rudiyanto et al., 2019; Fatchurrachman et al., 2022). Future studies could incorporate water proxies derived from more precise land cover data. This would help minimise the presence of mixed pixels in areas where rice fields and water bodies intersect.

Employing unsupervised classification with the grid in the LUCK-PALM method could result in artifacts along the boundaries of the grids (Amatulli et al., 2020). It is strongly influenced by the local spatial properties of the data (Tan et al., 2006) which may be disjointed with the neighbouring grid pattern. A grid cell in a moving window approach (Radoux et al., 2014) may be able to reduce this limitation, although it increases computation substantially.

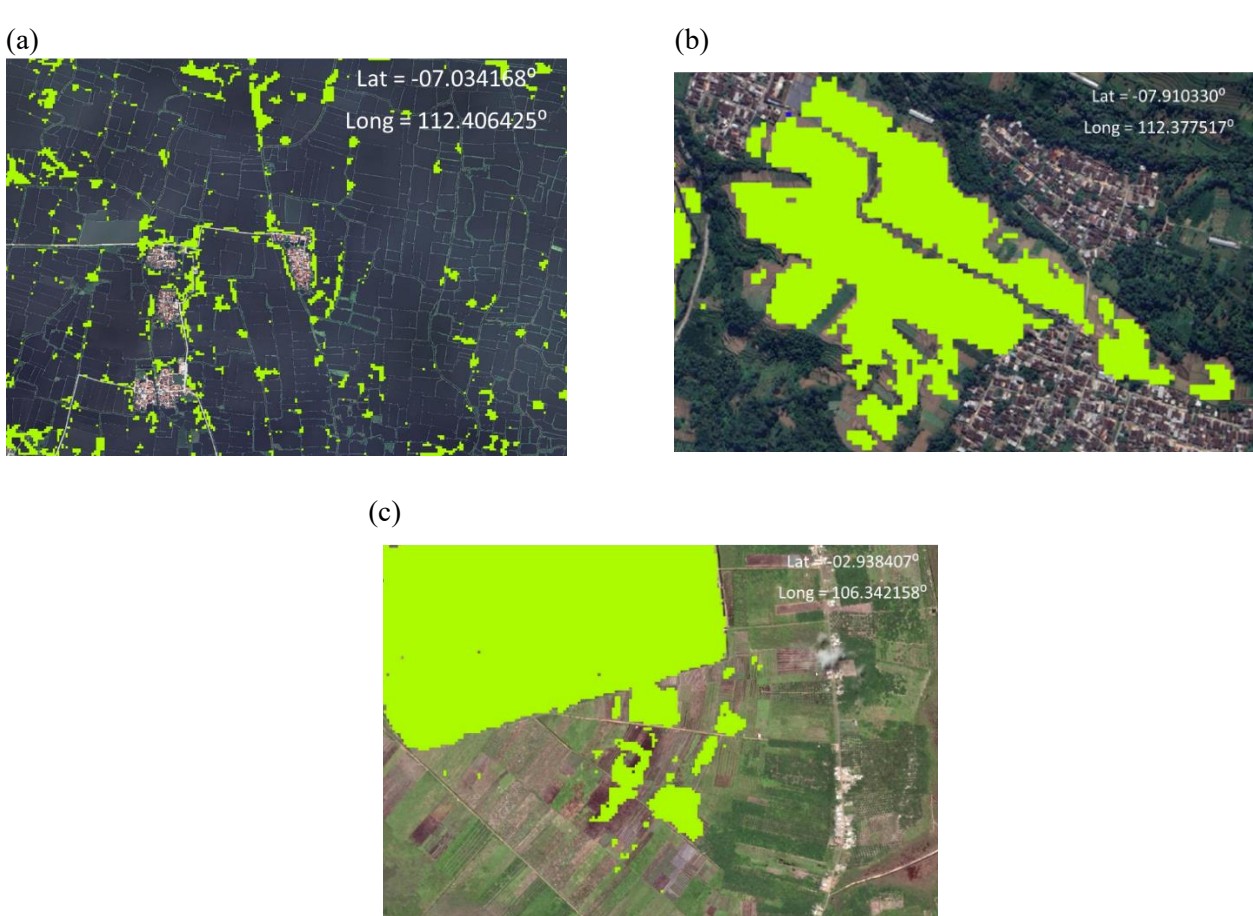

**Figure 18. Sources of classification error in paddy fields superimposed on © Google Earth. (a) Pond bunds or border areas**
**mistakenly detected as rice fields. (b) Mixed pixels resulting from the conversion of rice fields to horticultural crops. (c) Mixed pixels occurring at the intersection of rice and other crop classes. The green colour shows miss classification which is non-rice regions classified as rice.**

## 5 Conclusions

This study is the first to map and report rice cropping intensity and the harvested area across Southeast Asia at a spatial
resolution of 10 m (SEA-Rice-Ci10). We have developed a geospatial inventory of paddy rice parcels and rice cropping
intensity by integrating Sentinel-1 and 2 time-series data in a framework called namely LUCK-PALM, based on local
phenological expert interpretation. The spatial resolution of this database is 10 meters, which is, to our knowledge, the finest-
resolution and most accurate database of paddy rice in Southeast Asia. The distribution of paddy rice in our database was
highly correlated with government statistics, with an $R^2$ of 0.85 to 0.98. The paddy fields map from our study is more accurate
and has a higher level of detail than the existing products. Our method reduced the effects of mixed pixels and provided more
detailed spatial information than maps solely relying on optical and radar data. The method is effective in detecting rice
cropping intensity in diverse and complex regions like Southeast Asian countries. The data can be used for water security
management, greenhouse gas accounting, and forming government policies to address national food security and reduce
poverty.

Rice fields are also a significant source of methane, making the spatial data distribution of harvested rice in Southeast
Asia be used for greenhouse gas accounting valuable for these goals. As such, the results from this study have been used to
estimate methane emissions from rice cultivation in Southeast Asia as part of the Climate TRACE initiative
(https://climatetrace.org/). This data is freely downloadable to the public and can be used for greenhouse gas accounting in
regions that lack detailed rice cultivation and spatial resolution information needed for a country's emission inventory.

**Data availability**

The 10m paddy rice cropping intensity map for Southeast Asia, SEA-Rice-Ci10, is accessible on a public repository at the
following link: https://doi.org/10.5281/zenodo.10707621 (Frisa Irawan et al., 2024). The datasets utilize the EPSG:4326 spatial
reference system. The SEA-Rice-Ci10 map can be viewed from the Google Earth Engine App
(https://rudiyanto.users.earthengine.app/view/seariceci2021). The derived rice cultivation methane emissions estimates can be
accessed and downloaded from the Climate TRACE platform (https://climatetrace.org/).

**Supplement**

The supplement related to this article is available online.

**Author contribution**

Conceptualization: R, DF, BIS, BM, RMS, NCS, SGEG, SS, and AD; methodology: R and BM; software: FIG and R;
validation: FIG and R; formal analysis: FIG, F and R; investigation: FIG, F and R; resources: FIG, F and R; data curation:



FIG, F and R; writing—original draft preparation: FIG and R.; writing—review and editing: BM, R, SS, and AD; visualization: FIG. All authors have read and agreed to the published version of the manuscript.

**Competing interests**

The contact author has declared that none of the authors has any competing interests.

**Acknowledgements**

The authors would like to thank WattTime (Grant number: WattTime/2021/Vot 53416), the Climate TRACE coalition for their organizational support, and Climate TRACE funders- Al Gore, Generation Investment Management and Patrick J. McGovern Foundation. The computational aspect of this study was supported by The Group on Earth Observations (GEO) and Google Earth Engine (GEE) award.

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
