# Peer review of "SEA-Rice-Ci10: High-resolution Mapping of Rice Cropping Intensity and Harvested Area Across Southeast Asia using the Integration of Sentinel-1 and Sentinel-2 Data"

_Earth System Science Data, 2024_

## Author Comment (AC1)

Dear Editors and Reviewers,

We very much appreciate the constructive comments from the reviewers, which have helped improve our manuscript, "SEA-Rice-Ci10: High-resolution Mapping of Rice Cropping Intensity and Harvested Area Across Southeast Asia using the Integration of Sentinel-1 and Sentinel-2 Data" (MS No: essd-2024-90). Our detailed responses to the comments are included in the supplement with the following notes:

- The original review comments (in black)
- Our response on how the manuscript was revised (in red) and
- Revised paragraphs in the new manuscript (in blue)

Most or all suggestions are included in the revised manuscript. We are also submitting an annotated version of the revised manuscript.

Sincerely,

Rudiyanto, on behalf of all co-authors

Email: rudiyanto@umt.edu.my

**Citation**: https://doi.org/10.5194/essd-2024-90-RC1

**Reviewer #1:**

**RC1**: 'Comment on essd-2024-90', Anonymous Referee #1, 25 Apr 2024

The SEA-Rice-Ci10 study devised a novel approach called the Local Unsupervised Classification with Phenological Labeling Method (LUCK-PALM). This method aimed to accurately quantify and map rice cropping intensity and harvested areas across Southeast Asia from 2020 to 2021, with a 10-meter resolution using time-series inputs

from Sentinel-1 and Sentinel 2A/B imagery. The LUCK-PALM method is able to resolve the challenge due to the discrepancy in rice cropping calendar among the country and regions. Compared to agricultural statistics and existing rice maps, the study demonstrates its strength in retrieving field paddy details. The study, being open-source and featuring an open dataset, offers valuable insights into agricultural decision-making and management practices, such as methane emissions from rice cultivation, but the quality of the manuscript could be enhanced by addressing the following major and minor comments regarding methodology and content organization, etc. Please refer to the detailed feedback provided for further improvement.

**Response:** Thank you very much for the constructive comments. We will response to your comments per point below.

**Major comments:**

1. As shown in Fig.2 and Section 2.3.5, the "Google Map Very High Resolution and Street View" is the data source for exports to label rice and non-rice areas. Although examples and locations of such street view imagery are given in Fig.3, Fig.17 and Figure.S2/Table.S4, it would be good if more details about the street view collection for validations are provided. For example, how was each street view image acquired and how many validation points did the experts label in each grid or each country/province/district? Were those points selected randomly or the selection process was the same as the 2,000 random samples from each defined area?

**Response:** Thank you for your comments on clarifying the methodology used in this study. Our method is actually an extension of our method in our previous study, which had relatively high accuracy with a high Kappa coefficient of 0.92 for classifying rice and non-rice fields across Peninsular Malaysia, as we stated in Section 2.3, Lines 169-173. We have improved the description of the methodology especially the roles of VHRI and street view in the labelling process on section 2.3.5 Line 268 to 272. Here, we stressed that VHRI and street view images only provided information on rice and non-rice field classes, not cropping intensity. Cropping intensity is obtained from representative spectra of the cluster (Section 2.3.6).

We introduce the local unsupervised classification with phenological labelling (LUCK-PALM) method to produce a map of rice growing intensity (SEA-Rice-Ci10). This framework is the extension of our previous work (Fatchurachman et al., 2022) which used phenology-based approach by combining Sentinel-1 and Sentinel-2 time series data for mapping rice extent and cropping calendar across Peninsular Malaysia. They reported a high Kappa coefficient of 0.92 for classifying rice fields. In this extension, we included an additional index to detect soil water changes, Modified Normalized Difference Water Index (MNDWI) generated from Sentinel-2 data. The flowchart of this study is illustrated in Fig. 2, and each step is explained in the sections below:

and

Moreover, the identification of rice and non-rice fields was also assisted by overlaying the resulting cluster map on the very high-resolution image (VHRI) base map (i.e. satellite layer) in GEE and the Street View images in Google Maps. We manually checked the clusters with the VHRI and Street View images to verify the spectra profiles interpretation is correct in distinguishing rice fields from other crops (Rudiyanto et al., 2019; Zhang et al., 2020; Fatchurrachman et al., 2022).

2. The manuscript provided an assumption (line 219-220) that 25 to 30 clusters output from the unsupervised K-Means classification method would sufficiently represent the spectral data variations in each grid – with 2,000 points and 72 bands. Further explicit explanations or justifications for this assumption should be provided to offer readers a clearer understanding of its methods and enhance transferability.

**Response:** Thank you for the valuable comments. We added the justification on the sample size and number of clusters on Line 236-238. The sampling size and cluster number were based on a pre-experiment trial, considering both the result representativeness and computational time. A larger number of sample size would make the clustering inefficient and time consuming. We found that 25-30 clusters adequately represent the spectral data variations because the variation of imagery data has been reduced by applying non-crop mask as well as monthly value composite

in the imagery data. As we labelled or identified each cluster, we ended up grouping similar clusters to represent rice growing patterns.

"The selection of the number of sampling points and cluster numbers was based on trial assessments that balanced the representativeness of results and computational efficiency. We found that the chosen sample size and number of clusters sufficiently capture the variations in the spectral data."

3. The study trained local K-Means models for each grid using 2,000 randomly sampled points, and then assessed the accuracy of the produced maps with respect to agricultural statistics and existing products. However, descriptions about how the study applied the locally trained models in combination with expert knowledge to produce grid-wise maps as well as the compilation of maps into national/provincial/regency scales were missing from the manuscript. Also, as seen from the workflow of the study presented in Fig.2, the "Accuracy Assessment" was before the "Map of rice field extent and cropping intensity," which could potentially lead to the confusion that if the accuracy assessment was conducted with the 2,000 samples each grid or with the final mapping product.

**Response:** Thank you for your comments. We have updated the methodology section 2.3.3 to 2.3.6 as well as flowchart in Fig 2.

[Figure]

**Figure 2: Workflow of LUCK-PALM for mapping rice extent and cropping intensity using combined Sentinel-1 and Sentinel-2 time-series data.**

**2.3.4. Extracting representative spectra profiles**

To identify rice or non-rice groups, it is essential to obtain representative spectra of VH backscatter, NDVI, and MNDWI for each cluster generated by the K-means algorithm. To extract these representative spectra for each cluster, we randomly sampled 2,000 points within the defined area (i.e., for each grid) using the "sample ()" function in GEE and then computed the spatial median of each cluster using the "ee.Reducer.median()" function in GEE (Rudiyanto et al., 2019; Fatchurrachman et al., 2022). Figure 3 illustrates examples of representative spectral profiles of VH backscatter, NDVI, and MNDWI for clusters representing rice fields, water bodies, trees, and built-up areas.

**2.3.5 Expert labelling for identifying paddy rice fields and non-rice fields**

Based on the representative spectra profiles, expert labelling was conducted by visually inspecting each spectrum to identify clusters associated with rice fields and non-rice fields. Rice fields have a unique time series spectra profile in which the VH backscatter and NDVI values fluctuate seasonally and differ from other non-crop land uses (e.g., water, built-up, trees) as shown in Fig. 3. Spectral time series of non-cropping areas were relatively constant. The NDVI and VH backscatter coefficients of paddy rice change as it grows and matures. During rice transplanting or flooding phase, NDVI and VH backscatter coefficients have the lowest value while MNDWI reaches a maximum peak. The NDVI and VH backscatter coefficients rise after transplanting as the paddy rice grows and develops a peak at the heading stage (Davitt et al. 2020; Zhang et al., 2020b; Huang et al., 2023). After the rice harvest, the NDVI and VH backscatter coefficients decrease (Ramadhani et al., 2020; Fatchurrachman et al., 2022).

Next, we also differentiated between rice field and other crops. In Southeast Asian countries, rice is cultivated parallel with other crops such as sugarcane (Thailand, Indonesia, Philippines) and cassava (Thailand). Figure 4a shows an example of NDVI profiles to distinguish between rice and sugarcane and cassava in Thailand. The length of one season of rice is around 4 months for irrigated rice fields and around 7 months for rainfed rice fields, while both sugarcane and cassava have a longer season of around 10 months. In Indonesia, cash crops like maize and water melon are planted in rice fields during the dry season as reported by Rudiyanto et al. (2019). The NDVI and MNDWI profiles for a double rice cropping system followed by a cash crop is shown in Fig 4b. Rice season can be distinguished by examining the MNDWI signal peak during transplanting, which is higher in rice fields (0.01) compared to other crops (-0.20) due to standing water.

Moreover, the identification of rice and non-rice fields was also assisted by overlaying the resulting cluster map on the very high-resolution image (VHRI) base map (i.e. satellite layer) in GEE and the Street View images in Google Maps. We manually checked the clusters with the VHRI and Street View images to verify the spectra profiles interpretation is correct in distinguishing rice fields from other crops (Rudiyanto et al., 2019; Zhang et al., 2020; Fatchurrachman et al., 2022).

2.3.6 Expert labelling for identifying paddy rice cropping intensity

After identifying clusters as rice fields, we determined their cropping intensity based on the number of NDVI peaks (i.e., the number of rice seasons). Figure 5 illustrates the standard NDVI temporal profile for different paddy rice planting systems (single, double, and triple) on Java Island, Indonesia. In this labeling process, we manually labeled each cluster using the "remap()" function in GEE. We assigned the integer 0 to the non-rice class, and 1, 2, and 3 to single, double, and triple rice cropping intensities, respectively. After that the results were exported in the GeoTIFF raster file format. From these results, the total growing area and harvested area were calculated by the following formulas, respectively:

Total growing area = growing area with single season + growing area with double season + growing area with triple season $\hspace{3cm}$ (1)

Total harvested area = growing area with single season + (2 x growing area with double season) + (3 x growing area with triple season $\hspace{2cm}$ (2)

(a)

[Figure]

(b)

[Figure]

Figure 4. Representative spectra profiles for (a) NDVI of rice and other crops (sugarcane and cassava) and (b) NDVI and MNDWI for double rice season followed by cash crops

[Figure]

Figure 5. Representative spectra profiles of NDVI for single (blue lines), double (green lines) and triple (red lines) rice cropping intensities.

4.  The methods presented in the manuscript featured the workflow from Sentinel time series to rice / non-rice crop mapping (in terms of spatial distribution), but it appears that descriptions about how the authors retrieve the cropping intensity of rice (from time series-based spectral profiles?) were less elaborated in methods.

**Response:** Thank you for your suggestion, we further elaborated how to determine rice cropping intensity. We have added it in the new section 2.3.6.

**Minor comments:**

1.  In Fig.2, it is good to be concise in workflow illustrations, but how high is "Google Maps Very High Resolution" presented here? Also, the capitalization styles of words in Fig.2. could be more consistent.

**Response:** Thank you. Now we have revised using consistent words in Fig 2. Roles of are explained section 2.3.5 Line 268 to 272.

2. In Section 2.3.2 (line 192-195), what is the difference between using different landcovers from the WorldCover dataset to "filter out" non-croplands and using waterbody, tree, and built-up layers from the same dataset to "mask" non-cropland areas? The goal to facilitate computation and processing, as well as improving model performance is clear here, but the description of this step could be clearer.

**Response:** Thank you for your clarification. It is the same meaning as "masking". Thus to avoid redundancy, we removed "Waterbody, tree, and build-up layers from the WorldCover dataset were employed to mask non-cropland areas in Southeast Asia."

3. 3 appears to be coarser than other figures provided in the manuscript in terms of resolution.

**Response:** The resolution of Figure 3 has been improved.

4. 18 could have legends and classification accuracy and/or coefficients of determinant labeled on the map for each region.

**Response:** $R^2$ and RMSE from relationship between map products and statistical data for harvested area were added in Figure 18. We note that the harvested area for 20mRice-MSEAsia and NESEA-Rice 10m in these regions was calculated from the growing area with a single season (see Eq (2) in section 2.3.6), as rice fields in these regions have a single cropping intensity.

5. Line 642 has a typo: "penological mapping."

**Response:** Thank you, we have revised it Line 707.

**Citation**: https://doi.org/10.5194/essd-2024-90-RC1

---

## Author Comment (AC2)

Dear Editors and Reviewers,

We greatly appreciate the constructive comments provided by the reviewers, which have significantly improved our manuscript, "SEA-Rice-Ci10: High-resolution Mapping of Rice Cropping Intensity and Harvested Area Across Southeast Asia using the Integration of Sentinel-1 and Sentinel-2 Data" (MS No: essd-2024-90). Our detailed responses to these comments are included in the supplement, organized as follows:

- The original review comments (in black)
- Our response on how the manuscript was revised (in red) and
- Revised paragraphs in the new manuscript (in blue)

We have incorporated most or all of the reviewers' suggestions into the revised manuscript. Additionally, we are submitting an annotated version of the revised manuscript for your reference.

Sincerely,

Rudiyanto, on behalf of all co-authors

Email: rudiyanto@umt.edu.my

Citation: https://doi.org/10.5194/essd-2024-90-RC2

**Reviewer #2:**

**RC2: 'Comment on essd-2024-90', Anonymous Referee #2, 30 Apr 2024**

This manuscript identifies the distribution of different maturation types of rice in Southeast Asia at a 10m resolution, generating the up-to-date maps of planting intensity and area in Southeast Asia, which is interesting. However, there are some

non-negligible flaws in the paper. The study period is too short, covering only 2020-2021, so the data used for validation and comparison do not align with the study period perfectly. Comparisons with existing data lack a unified standard, and the description of the core methodology is not clear enough. Below are specific comments:

**Response:** Thank you very much for your valuable comments. We will response your comments per point below.

Comment 1: The results of $R^2$ through the entire MS should be checked. Section 2.3.6 illustrated that $R^2$ is determined by mapping results and statistics (or areas from existing datasets), however, the $R^2$ values presented in the results seem to denote the goodness of fit, that is, the correlation with fitting lines. For instance, the declared consistency $R^2$ for Fig.15(b) is 0.86, which is more like the correlation degree with the fitting line, not with the NESEA. RMSE values should also be confirmed.

**Response:** Thank you for your clarification. $R^2$ value is the goodness of fit of linear regression and it is identical to $R^2$ (Eq. 3). We have checked it as well as RMSE values. In case Fig 15b. It has a relatively high $R^{2;}$ however, its slope is relatively high (1.75) which indicates that the estimated area by NESEA rice is lower than our results in this study.

Comment 2: Lines 22-23: The statement, "This paper presents the first detailed study of rice cropping intensity and harvested areas throughout Southeast Asia," requires further qualification in terms of temporal and spatial resolution. This is because the study is not **the first of its kind when considering coarser spatial resolutions.**

**Response:** Thank you. We added "at a 10-meter spatial resolution" Line 23.

This paper presents the first detailed study of rice cropping intensity and harvested areas throughout Southeast Asia at a 10-meter spatial resolution.

Comment 3: Lines 56-57: In many regions of China, rice fields undergo two harvests per season.

**Response:** Thank you for your clarification. We revised it as "the northeast China (Heilongjiang region)" line 57.

Summing individual rice fields in Southeast Asia does not equate to the total harvested area, unlike in temperate regions such as northeast China (Heilongjiang region), India, the USA, Italy, and Japan where rice paddies typically produce one harvest a season.

Comment 4: Line 76-77: What exactly does "requiring in-depth expertise for labelling time series of vegetation indices for rice growth stages" mean?

**Response:** Thank you. We have clarified line 78.

Conversely, expert phenology-based classifiers are straightforward, albeit requiring in-depth expertise in differentiating rice and non-rice classes and labelling time series of vegetation indices for rice growth stages (Rudiyanto et al., 2019; Han et al., 2021; Fatchurrachman et al., 2022)

Comment 5: Why did the study only consider 2020-2021? Both sentinel-1 and sentinel-2 data from 2017 onwards are available. The datasets available for comparison extend only up to 2019; however, this study did not generate a map for that year.

**Response:** Thank you for your inquiry. The study focused on the years 2020-2021 primarily due to our data processing timeline, which began in the middle of 2022. This timeframe allowed us to work with the most current and complete dataset available, as data for 2021 was the latest year that had been fully processed at the time of our analysis.

Additionally, the decision to use consecutive years (2020 and 2021) was based on the phenology approach employed in our study. This approach requires the observation of the complete rice growth stages, given that rice is typically planted towards the end

of one year (October, November, December) and harvested in the following year (January, February, March) in certain regions such as Kedah, Malaysia; the northern region of West Java, Indonesia; and parts of South and North Vietnam. Using consecutive years ensured that we captured the necessary phenological information for our analysis. We have discussed it Line 120 to 125.

Two consecutive years of data were used because this study's phenology approach necessitates observing the entire rice growth cycle. In certain regions such as Kedah, Malaysia; the northern region of West Java, Indonesia; and parts of South and North Vietnam, rice is planted towards the end of the year (October, November, December) and harvested in the subsequent year (January, February, March). This timeframe ensures the comprehensive assessment of rice growth stages.

Comment 6: Table 1 does not need to be listed separately.

**Response:** Thank you for your suggestion, we have updated Table 1.

Comment 7: In Table 2, some links to the statistical data are not functioning. Please check all links provided for accessing statistical data to ensure they directly lead to the specific webpage where the data can be accessed, not to the website's homepage.

**Response:** Thank you for the comments. We have updated it. Now all links in Table 2 are functioning.

Comment 8: The flow and details of the core method were confused.

(1) What is the scope of the unsupervised classification? Does it only include the crop layer, or the entire area? I was confused about whether to group the crop layers into rice and non-rice clusters, or group the whole area into 25 ~ 30 clusters.

**Response:** Thank you for your clarification. We masked out non-crop area using WorldCover product (section 2.3.2). Thus, we only used areas under cropping based on WorldCover products for the unsupervised classification (section 2.3.3).

(2) How many labels are available for each cluster? Besides rice, water, trees, and built-up areas, there should be some other categories. Do the available labels cover all the major land types?

**Response:** Thank you for your clarification. In this labelling, we used integer 0 for non-rice class and 1, 2 and 3 for single, double, and triple rice cropping intensities, respectively. We explained it in section 2.3.4, 2.3.5 and 2.3.6.

2.3.4. Extracting representative spectra profiles

To identify rice or non-rice groups, it is essential to obtain representative spectra of VH backscatter, NDVI, and MNDWI for each cluster generated by the K-means algorithm. To extract these representative spectra for each cluster, we randomly sampled 2,000 points within the defined area (i.e., for each grid) using the "sample ()" function in GEE and then computed the spatial median of each cluster using the "ee.Reducer.median()" function in GEE (Rudiyanto et al., 2019; Fatchurrachman et al., 2022). Figure 3 illustrates examples of representative spectral profiles of VH backscatter, NDVI, and MNDWI for clusters representing rice fields, water bodies, trees, and built-up areas.

2.3.5 Expert labelling for identifying paddy rice fields and non-rice fields

Based on the representative spectra profiles, expert labelling was conducted by visually inspecting each spectrum to identify clusters associated with rice fields and non-rice fields. Rice fields have a unique time series spectra profile in which the VH backscatter and NDVI values fluctuate seasonally and differ from other non-crop land uses (e.g., water, built-up, trees) as shown in Fig. 3. Spectral time series of non-cropping areas were relatively constant. The NDVI and VH backscatter coefficients of

paddy rice change as it grows and matures. During rice transplanting or flooding phase, NDVI and VH backscatter coefficients have the lowest value while MNDWI reaches a maximum peak. The NDVI and VH backscatter coefficients rise after transplanting as the paddy rice grows and develops a peak at the heading stage (Davitt et al. 2020; Zhang et al., 2020b; Huang et al., 2023). After the rice harvest, the NDVI and VH backscatter coefficients decrease (Ramadhani et al., 2020; Fatchurrachman et al., 2022).

Next, we also differentiated between rice field and other crops. In Southeast Asian countries, rice is cultivated parallel with other crops such as sugarcane (Thailand, Indonesia, Philippines) and cassava (Thailand). Figure 4a shows an example of NDVI profiles to distinguish between rice and sugarcane and cassava in Thailand. The length of one season of rice is around 4 months for irrigated rice fields and around 7 months for rainfed rice fields, while both sugarcane and cassava have a longer season of around 10 months. In Indonesia, cash crops like maize and water melon are planted in rice fields during the dry season as reported by Rudiyanto et al. (2019). The NDVI and MNDWI profiles for a double rice cropping system followed by a cash crop is shown in Fig 4b. Rice season can be distinguished by examining the MNDWI signal peak during transplanting, which is higher in rice fields (0.01) compared to other crops (-0.20) due to standing water.

Moreover, the identification of rice and non-rice fields was also assisted by overlaying the resulting cluster map on the very high-resolution image (VHRI) base map (i.e. satellite layer) in GEE and the Street View images in Google Maps. We manually checked the clusters with the VHRI and Street View images to verify the spectra profiles interpretation is correct in distinguishing rice fields from other crops (Rudiyanto et al., 2019; Zhang et al., 2020; Fatchurrachman et al., 2022).

2.3.6 Expert labelling for identifying paddy rice cropping intensity

After identifying clusters as rice fields, we determined their cropping intensity based on the number of NDVI peaks (i.e., the number of rice seasons). Figure 5 illustrates the standard NDVI temporal profile for different paddy rice planting systems (single, double, and triple) on Java Island, Indonesia. In this labelling process, we manually labelled each cluster using the "remap()" function in GEE. We assigned the integer 0

to the non-rice class, and 1, 2, and 3 to single, double, and triple rice cropping intensities, respectively. After that the results were exported in the GeoTIFF raster file format. From these results, the total growing area and harvested area were calculated by the following formulas, respectively:

Total growing area = growing area with single season + growing area with double season + growing area with triple season                                                 (1)

Total harvested area = growing area with single season + (2 x growing area with double season) + (3 x growing area with triple season                                  (2)

(a)

[Figure]

(b)

[Figure]

Figure 4. Representative spectra profiles for (a) NDVI of rice and other crops (sugarcane and cassava) and (b) NDVI and MNDWI for double rice season followed by cash crops

[Figure]

Figure 5. Representative spectra profiles of NDVI for single (blue lines), double (green lines) and triple (red lines) rice cropping intensities.

(3) Following on question (2), how to distinguish rice from other crops?

**Response:** Thank you for your clarification. We have explained in detail in section 2.3.5 line 260 to 268.

(4) If not binary classification, how to determine all rice pixels or clusters according to the representative profiles? Are there any quantitative standards?

**Response:** Thank you for your clarification. We explained the process in section 2.3.4, 2.3.5 and 2.3.6. We also note that rice spectra characteristics highly vary from place to place due to climate, farm management practices, etc as shown by the representative spectra profiles in Fig 6 to 14. So, it is difficult to get a set of standards thus we used expert based observation by visual inspection of representative spectra profiles which assisted and complemented by VHRI Google Maps and Street View.

(5) For the clustering results in hundreds of grids, did the author manually label each cluster?

**Response:** Thank you for your clarification. We explained in section 2.3.5 and 2.3.6. Yes, we manually labelled each cluster using "remap()" function in GEE.

After identifying clusters as rice fields, we determined their cropping intensity based on the number of NDVI peaks (i.e., the number of rice seasons). Figure 5 illustrates the standard NDVI temporal profile for different paddy rice planting systems (single, double, and triple) on Java Island, Indonesia. In this labelling process, we manually labelled each cluster using the "remap()" function in GEE. We assigned the integer 0 to the non-rice class, and 1, 2, and 3 to single, double, and triple rice cropping intensities, respectively.

Comment 9: Why does the clustering utilize a two-year continuous time series as input features rather than a one-year time series? To which year does the rice mapping area pertain when using this clustering method based on two-year time series data? I am concerned that this approach may overlook variations in rice cultivation area across different years, which could affect the accuracy of subsequent comparisons between rice maps and statistical data.

**Response:** Thank you we have responded in comment-5.

Comment 10: To my knowledge, even with monthly composites, the Sentinel-2 time series data exhibits significant gaps in Southeast Asia. How do the authors address the issue of these data gaps?

**Response:** Thank you for raising this important point. When either Sentinel-1 or Sentinel-2 data was unavailable for a particular month, we made the decision to exclude that specific month from our analysis. However, it's important to note that even

if we have a gap of Sentinel-2 data for a given month, we still retained the corresponding data from Sentinel-1 for that timeframe. This approach helped mitigate the impact of missing data and ensured that our analysis was as comprehensive and accurate as possible given the available information.

Comment 11: The manuscript presents numerous curves of NDVI, VH, and MNDWI for rice. It is necessary to clearly specify the samples these curves are based on.

**Response:** Thank you for comments. We have explained in section 2.3.4 Extracting representative spectra profiles

Comment 12: Are there any directly available sources for Statistics rice growing area in Table 3? Or is it calculated indirectly by using the harvested area of different cropping patterns?

**Response:** We have provided the data link in Table 2 corresponding to the data specification column, indicating whether it is "growing area" or "harvested area."

Comment 13: The statistics referenced in Section 3.2 include data from years other than 2020 and 2021. How to address the discrepancies between the mapping year and the statistical year?

**Response:** Thank you for bring up this important point. It is indeed common for statistical data to become available with a delay of one or two years. Despite the temporal difference between the mapping year and the statistical data years, comparisons remain reliable due to the relative stability of rice fields in the region. Furthermore, we have not observed any extreme conditions such as widespread drought affecting a large area within the region.

Comment 14: Table 5 stated that the consistency of Timor-Leste with the statistical area is compared using "growing area" statistics. Why are there no consistency results for NESEA?

**Response:** We apologize for the oversight and confusion regarding the comparison presented in Table 5. Upon careful review, we realized that the information related to Timor-Leste was mistakenly included. the value is correlation between this study vs. NESEA. Thus, we removed it. We did not find the area by either adm 1 or 2.

Comment 15: Is Figure 15 a comparison based on "growing area" or "harvested area"? How to determine the relative accuracy of areas that differ greatly from existing datasets? Without a unified standard of reference, such comparisons are meaningless because it is uncertain which is more accurate.

**Response:** Figure 15 presents rice growing area or parcel area. We note that two existing products (NESEA-Rice10 and MSEA rice 20) only provide growing area. Thus, we only compared in term of growing area, as we could not compare our harvested area results from this study with theirs.

Comment 16: Similarly, the spatial comparison lacks a real reference and is not based on the same year products.

**Response:** we have responded in comment 13.

Comment 17: To my understanding, the NESEA-Rice10 rice map is generated based on flood signals, primarily focusing on irrigated rice fields and scarcely including rainfed types. In this study, according to the authors, the rice map includes rainfed rice (although it remains unclear how the authors determine which category within the clustering represents rice), which may account for the main difference in area statistics between NESEA-Rice10 and this work. The authors need to clearly describe the differences in rice types included in the different data products when making comparisons.

**Response:** Thank you for your valuable comments. The types of data used are shown in the "Data specification" column in Table 2, indicating whether it is "cropping area" or "harvested area." These data types are compared in the Fig 17 (growing area). We have discussed limitation of NESEA-Rice in section 4.2 line 753 to 757.

Comment 18: The study lacks individual comparisons of areas with different rice cropping intensities. Therefore, further validation of the accuracy of rice intensity measurements is necessary.

**Response:** The method for determining rice cropping intensity is explained clearly in section 2.3.6. Cropping intensities were observed visually based on the number of phenological peaks in the NDVI representative spectra. We have checked from all generated spectra profiles in section 2.3.4 and 2.3.5. Existing products are at a coarse resolution of 500 m:

https://www.sciencedirect.com/science/article/pii/S0308521X22000737

and 0.5° (approximately 55 km):

https://essd.copernicus.org/preprints/essd-2023-283/.

Therefore, we did not compare our data with these products.

Comment 19: How is the rice map validated using Google Street View data? There is a lack of more extensive validation based on sample points.

**Response:** Thank you for your question. We have discussed how we used Google Street View to verify our map in section 2.3.5. Here, we stressed that VHRI and street view images only provided information on rice and non-rice field classes, not cropping intensity. Cropping intensity is obtained from representative spectra of the cluster (Section 2.3.6).

Citation: https://doi.org/10.5194/essd-2024-90-RC2